# Learning Distributionally Robust Models at Scale via Composite Optimization

**Farzin Haddadpour**
Yale Institute for Network Science
Yale University
farzin.haddadpour@yale.edu

**Mohammad Mahdi Kamani**
Wyze Labs Inc.
mmkamani@alumni.psu.edu

**Mehrdad Mahdavi**
Department of Computer Science & Engineering
The Pennsylvania State University
mzm616@psu.edu

**Amin Karbasi**
Yale Institute for Network Science
Yale University
amin.karbasi@yale.edu

## Abstract

To train machine learning models that are robust to distribution shifts in the data, distributionally robust optimization (DRO) has been proven very effective. However, the existing approaches to learning a distributionally robust model either require solving complex optimization problems such as semidefinite programming or a first-order method whose convergence scales linearly with the number of data samples– which hinders their scalability to large datasets. In this paper, we show how different variants of DRO are simply instances of a finite-sum composite optimization for which we provide scalable methods. We also provide empirical results that demonstrate the effectiveness of our proposed algorithm with respect to the prior art in order to learn robust models from very large datasets.

## 1 Introduction

Conventional machine learning problem aims at learning a model based on the assumption that training data and test data come from same data distribution. However, this assumption may not hold in various practical learning problems where there is label shift (Zhang et al., 2020a), distribution shift (Sagawa et al., 2019), fairness constraints (Hashimoto et al., 2018), and adversarial examples (Sinha et al., 2017), to name a few. Distributionally robust optimization (DRO), which has recently attracted remarkable attention from the machine learning community, is a common approach to deal with the aforementioned uncertainties (Chen et al., 2017; Duchi & Namkoong, 2016; Rahimian & Mehrotra, 2019). Defining the empirical distribution of the training data of size $m$ by $\hat{\mathbb{P}}_m \triangleq \frac{1}{m} \sum_{i=1}^{m} \delta_{\hat{\xi}_i}$ where $\delta$ is the Dirac delta function, the goal of DRO is to solve the following optimization problem

$$\inf_{\boldsymbol{x}} \left[ \Psi(\boldsymbol{x}) \triangleq \sup_{\xi \in Q} \mathbb{E}_Q \left[ \ell(\boldsymbol{x}; \xi) \right] \right], \tag{1}$$

where $\xi$ is a data sample randomly drawn from distribution $Q$, $\ell(\boldsymbol{x}; \xi)$ is the corresponding loss function and $\mathbb{E}_Q \left[ \ell(\boldsymbol{x}, \xi) \right]$ is the expected loss over distribution $Q$ which belongs to uncertainty set $\mathcal{U}_m$. The uncertainty set $\mathcal{U}_m$ is defined as $\mathcal{U}_m \triangleq \{Q : d(Q, \hat{\mathbb{P}}_m) \leq \epsilon\}$ indicates the ball of a distribution with center $\hat{\mathbb{P}}_m$ and also $d(P, Q)$ is a distance measure between probability distribution $P$ and $Q$. We note this uncertainty set captures the distribution shift hence Eq. (1) minimizes the worse data distribution. Prior studies (Ben-Tal et al., 2013; Bertsimas et al., 2018; Blanchet et al., 2019; Esfahani & Kuhn, 2018; Pourbabaee, 2020) considered different uncertainty sets (see Definition 3.1 in Esfahani & Kuhn (2018)) for which they proposed equivalent reformulations of Eq. (1) based on the specific choice of $\mathcal{U}_m$.

To solve the above min-max optimization problems, majority of prior studies heavily rely on either semidefinite programming (Esfahani & Kuhn, 2018) or stochastic primal-dual methods both for convex (Deng et al., 2021; Nemirovski et al., 2009; Juditsky et al., 2011; Yan et al., 2019; 2020; Namkoong & Duchi, 2016) and non-convex (deep learning) objectives (Yan et al., 2020). While

primal-dual methods can be used as an approach to solve min-max optimization problems, it suffers from a few downsides. First and foremost, they need to store a probability distribution of constrained violation of dimension $m$ corresponding to dual variables. Additionally, available primal-dual methods often demand data sampling that corresponds to the probability distribution over $m$ data samples which introduces additional cost over uniform sampling. Finally, while majority of prior studies are limited to DRO problems with convex objectives, establishing tight convergence rate for DRO problems with penalty with non-convex objectives is still lacking.

To overcome these issues, we consider three different reformulations of Eq. (1), corresponding to three different choices of uncertainty sets $\mathcal{U}_m$ namely, *(1) DRO with Wasserstein metrics*, *(2) DRO with $\chi^2$ divergence metrics*, and *(3) DRO with regularized entropy metrics* (also known as KL) and show in Section 2 that all aforementioned DRO notions are indeed different instances of a *deterministic composite optimization* and can be solved by reducing to an instances of the following problem:

$$\min_{\boldsymbol{x}} \left[ \Psi(\boldsymbol{x}) \triangleq r(\boldsymbol{x}) + \frac{1}{m} \sum\nolimits_{i=1}^{m} h_i(\boldsymbol{x}) + f\left( \frac{1}{m} \sum\nolimits_{i=1}^{m} g_i(\boldsymbol{x}) \right) \right], \qquad (2)$$

where we suppose $r(\boldsymbol{x})$ is convex and a relatively simple function, $f(\boldsymbol{x}) : \mathbb{R}^p \to \mathbb{R}$ and $h_i(\boldsymbol{x}) : \mathbb{R}^d \to \mathbb{R}$ for $1 \le i \le m$ are scalar-valued functions, and $g_i(\boldsymbol{x}) : \mathbb{R}^d \to \mathbb{R}^p$ for $1 \le i \le m$ are vector-valued functions. On the road to solve problem (2) at scale, we also develop a novel algorithm for heavily constrained optimization problems (Narasimhan et al., 2020b; Wang & Bertsekas, 2015; 2016) that rather surprisingly invokes a *single* projection through the course of optimization. This algorithm is of independent interest and addresses the scalability issues raised in applications such as fairness (Donini et al., 2018; Zafar et al., 2019).

We summarize the main **contributions** of our paper below:

- We provide a large-scale analysis of DRO with Wasserstein distance and heavily constrained reformulation when the objective function is strongly-convex. Our result relies on a novel mini-batch constraint sampling for handling heavily-constrained optimization problems. As summarized in Table 1, our convergence analysis improves the state-of-the-art both in terms of the dependence on the convergence error $\epsilon$ as well as the number of constraints $m$.
- We represent a large-scale analysis of DRO with non-convex objectives and $\chi^2$ or KL divergences and propose a distributed varaint to further improve scalability of DRO problems.
- We verify our theoretical results through various extensive experiments on different datasets. In particular, we show that our proposed method outperforms recent methods in DRO for heavily constrained problems with a great reduction in time complexity over them.

The proofs of all the theorems are provided in the **appendix**.

## 1.1 RELATED WORK

**DRO and connections to heavily constrained optimization.** As mentioned earlier, DRO has many different formulations, depending on the divergence metrics used (e.g., Wasserstein , $\chi^2$ or KL). While Duchi & Namkoong (2021); Namkoong & Duchi (2016); Shapiro (2017) consider constrained or penalized DRO formulation, Levy et al. (2020); Sinha et al. (2017) formulate the underlying optimization problem as unconstrained. One of the contributions of our paper is to provide a unifying framework through the language of composite optimization and treat all these variants similarly.

In particular, when the objective function is convex, Levy et al. (2020) recently proposed scalable algorithms for different variants of the DRO problems with , e.g., $\chi^2$ or KL divergence metrics. Our unifying approach readily extends those results to the more challenging non-convex setting for which we are unaware of any prior work with convergence guarantees (for instance, Hashimoto et al. (2018) studied DRO with $\chi^2$-divergence but did not provide any convergence guarantee). Similarly, Esfahani & Kuhn (2018); Kuhn et al. (2019) formulated DRO with Wasserstein distance as an instance of constrained optimization. Notably, they require ti impose one constraint per training data point and to solve such a constrained problem they proposed a semi-definite program. Even though the formulation is very novel, it cannot scale. We, in contrast, consider such a heavily constrained optimization as an instance of a composite optimization for which we provide a scalable solution. What is rather surprising about our method is that it only checks a batch of constraints per iteration, inspired by Cotter et al. (2016), and performs a single projection at the final stage of the algorithm in order to provide an $\epsilon$-optimal solution in the case of strongly convex objectives. Moreover, in

contrast to Cotter et al. (2016), we do not keep a probability distribution over the set of constraints. We should also remark that our convergence guarantees achieve the known lower bounds in terms of accuracy $\epsilon$ and the number of constraints $m$. Finally, we should highlight the difference of our algorithm and Frank-Wolfe (FW) (Frank et al., 1956; Jaggi, 2013; Zhang et al., 2020b). While FW does not require a projection oracle, it performs a linear program over the set of constraints at each iteration. In contrast, our heavily- constrained optimization solution performs a single projection without the overhead of running a linear program at each iteration.

**Stochastic composite optimization.** The general stochastic composite optimization $\min_{\boldsymbol{x}} \left[ \Psi(\boldsymbol{x}) \triangleq r(\boldsymbol{x}) + f \left( \mathbb{E}_\xi \left[ g_\xi(\boldsymbol{x}) \right] \right) \right]$ has recently received a lot of attentions (Qi et al., 2020b;a; Wang et al., 2017; Kalogerias & Powell, 2019). Our reformulation of DRO variants is a finite-sum instance of this general problem. More concretely, Huo et al. (2018); Lian et al. (2017); Zhang & Xiao (2019a) aimed to solve the following finite-sum problem $\min_{\boldsymbol{x}} \left[ \Psi(\boldsymbol{x}) \triangleq r(\boldsymbol{x}) + \frac{1}{n} \sum_{j=1}^n f_j \left( \frac{1}{m} \sum_{i=1}^m g_i(\boldsymbol{x}) \right) \right]$, using SVRG or SAGA (Defazio et al., 2014). In contrast, our proposed algorithm is inspired by Zhang & Xiao (2019a) and generalizes their method to the case where the extra terms $h_i(\boldsymbol{x})$ in Eq. (2) are non-zero. We should also note that Qi et al. (2020a) proposed a similar idea in the context of online learning for DRO problems with KL divergence. Our work in contrast provides guarantees for DRO with both constraints or penalty terms.

## 2 DRO VIA FINITE-SUM COMPOSITE OPTIMIZATION

In this section, we discuss in detail how a finite-sum composite optimization (2) can unify various notions of distributionally robust learning, where some of which rely on heavily constrained optimization subroutines. While much research effort has been devoted to develop a specialized algorithm for each notion, our reduction paves the way to developing a scalable algorithm, discussed in Section 3.

**DRO with Wasserstein distance.** An equivalent and tractable reformulation of Eq. (1) is provided in Esfahani & Kuhn (2018); Kuhn et al. (2019), which can be regarded as a heavily constrained optimization problem as follows:

$$\min_{\boldsymbol{x}} r(\boldsymbol{x}) \triangleq \frac{1}{m} \sum_{i=1}^m f_i(\boldsymbol{x}) \qquad \text{subject to} \qquad \tilde{g}_i(\boldsymbol{x}) \leq 0, \, \forall i \in [m]. \tag{3}$$

where $\tilde{g}_i(\boldsymbol{x})$ are functions related to loss function as well as slack variables (please see Appendix A for more details). Naively solving optimization problem (3) suffers from the computational complexity due to the large number of constraints $m$. To efficiently solve the optimization problem (3), inspired by Mahdavi et al. (2012) and Cotter et al. (2016), we pursue the smoothed constrained reduction approach and introduce the augmented optimization problem (see Appendix B) of the form $\min_{\boldsymbol{x}} \Psi(\boldsymbol{x}) \triangleq [r(\boldsymbol{x}) + \gamma \ln (g(\boldsymbol{x}))]$ where $g_i(\boldsymbol{x}) \triangleq \exp\left( \frac{\alpha \tilde{g}_i(\boldsymbol{x})}{\gamma} \right)$ and $g(\boldsymbol{x}) = \frac{1}{m+1} [1 + \sum_{i=1}^m g_i(\boldsymbol{x})]$. We can see that this optimization problem is a special case of the optimization problem Eq. (2) where $r(\boldsymbol{x}) = f(\boldsymbol{x})$, $f(\frac{1}{m} \sum_{i=1}^m g_i(\boldsymbol{x})) = \gamma \ln g(\boldsymbol{x})$, and $h(\boldsymbol{x}) = 0$. In contrast to Cotter et al. (2016) that requires an extra storage cost of probability distribution of dimension $m$, and relatively poor convergence rate in terms of $m$ and accuracy $\epsilon$, we propose an algorithm that simply checks a batch of constraints and achieves the optimum dependency in terms of $m$ and $\epsilon$.

**DRO with $\chi^2$-divergence.** The second type of DRO problem we consider utilizes the $\chi^2$-divergence metric as follows:

$$\min_{\boldsymbol{x}} \quad \max_{0 \leq p_i \leq 1, \sum_{i=1}^m p_i = 1} \sum_{i=1}^m p_i f_i(\boldsymbol{x}_i) - \gamma D_{\chi^2}(\boldsymbol{p}). \tag{4}$$

where the $\chi^2$ divergence is defined as the distance between the uniform distribution and an arbitrary probability distribution $\boldsymbol{p}$, i.e., $D_{\chi^2}(\boldsymbol{p}) \triangleq \frac{m}{2} \sum_{i=1}^m \left( p_i - \frac{1}{m} \right)^2$. Levy et al. (2020) studied this problem only for the case of convex objectives. In this paper, we allow objective functions $f_i$ for $1 \leq i \leq m$ to be both non-convex or strongly-convex. The following claim derives the equivalent finite-sum composite optimization.

| Reference | DRO type | # Constraint/Sample Checks to achieve $\epsilon$ error | objective |
|---|---|---|---|
| Midtouch Cotter et al. (2016) | Wasserstein | $O\left(\frac{\ln m}{\epsilon} + \frac{m^{1.5}(\ln m)^{1.5}}{\epsilon^{\frac{3}{4}}} + \frac{m(\ln m)^{2/3}}{\epsilon^{\frac{2}{3}}} + \frac{m^2\log m}{\sqrt{\epsilon}}\right)$ | Strongly convex |
| SEVR Yu et al. (2021) | General Linearized Wasserstein | $O\left(m\ln\left(\frac{1}{\epsilon}\right) + \frac{D_L}{\epsilon} + \frac{(\ell+\kappa+1)^2 D_u^4}{\epsilon^2}\right)$ | Strongly convex-concave min-max |
| **Theorem 4.1** | **Wasserstein** | $O\left((m + \kappa\sqrt{m})\ln\frac{1}{\epsilon}\right)$ | **Optimally strongly convex** |
| **Theorem 4.3** | $\chi^2$ **or KL** | $\tilde{O}\left(\min\{\frac{\sqrt{m}}{\epsilon}, \frac{1}{\epsilon^{1.5}}\}\right)$ | **Non-convex** |
| **Theorem D.1** | $\chi^2$ **or KL** | $O\left((m + \kappa\sqrt{m})\ln\frac{1}{\epsilon}\right)$ | **Optimally strongly convex** |

Table 1: Comparison of our results with prior approached. All three approaches are using variance reduction techniques. $D_u$ and $D_L$ respectively denotes the upper bound on the distance of initial model from optimal model and initial optimality gap. Please see Yu et al. (2021) for more details. Finally, we note that while Midtouch approach in Cotter et al. (2016) requires additional storage of probability distribution of dimension $m$, our approach does not.

**Claim 2.1.** *The optimization problem (4) is equivalent to the following composite problem:*

$$\min_{\boldsymbol{x}} \quad \left[\Psi(\boldsymbol{x}) \triangleq 1 - \frac{1}{2\gamma m}\sum_{i=1}^{m}[f_i(\boldsymbol{x})]^2 + \frac{1}{2\gamma}\left[\frac{1}{m}\sum_{i=1}^{m}f_i(\boldsymbol{x})\right]^2\right] \tag{5}$$

We note that the optimization problem (5) fits into the formulation of finite-sum composite optimization (2) by choosing $r(\boldsymbol{x}) = 1$, $h(\boldsymbol{x}) = \frac{1}{m}\sum_{i=1}^{m} -\frac{(f_i(\boldsymbol{x}))^2}{2\gamma}$ and $f(g(\boldsymbol{x})) = \frac{1}{2\gamma}\left[\frac{1}{m}\sum_{i=1}^{m}f_i(\boldsymbol{x})\right]^2$ with $h_i(\boldsymbol{x}) = -\frac{f_i^2(\boldsymbol{x})}{2\gamma}$, $g_i(\boldsymbol{x}) = f_i(\boldsymbol{x})$ and $f(x) = \frac{x^2}{2\gamma}$.

**DRO with KL divergence.** Finally, for DRO with KL-divergence, usually considered in online settings (Qi et al., 2020a), we consider solving the following optimization problem:

$$\min_{\boldsymbol{x}} \quad \max_{0 \leq p_i \leq 1, \sum_{i=1}^{m} p_i = 1} \left[\sum_{i=1}^{m} p_i f_i(\boldsymbol{x}_i) + \gamma H(p_1, \ldots, p_m)\right], \tag{6}$$

where $H(p_1, \ldots, p_m) = -\sum_{i=1}^{m} p_i \log p_i$ is the entropy function. To solve problem (6), it is straightforward to convert it to the following equivalent stochastic composite optimization problem:

$$\min_{\boldsymbol{x}} \quad \left[\Psi(\boldsymbol{x}) \triangleq \ln\left(\frac{1}{m}\sum_{i=1}^{m}\exp\left(\frac{f_i(\boldsymbol{x})}{\gamma}\right)\right)\right]. \tag{7}$$

As it can be seen, the optimization problem (7) fits into the composite optimization (2) by choosing $r(\boldsymbol{x}) = h(\boldsymbol{x}) = 0$ and $f(g(\boldsymbol{x})) = \ln\left(\frac{1}{m}\sum_{i=1}^{m}\exp\left(\frac{f_i(\boldsymbol{x})}{\gamma}\right)\right)$.

## 3  OUR PROPOSED ALGORITHM

Having reduced the different notions of DRO to an instance of the composite optimization, in this section we describe our scalable approach for minimizing the objective $\Psi(\boldsymbol{x}) = r(\boldsymbol{x}) + \Phi(\boldsymbol{x})$ where $\Phi(\boldsymbol{x}) = \frac{1}{m}\sum_{i=1}^{m} h_i(\boldsymbol{x}) + f(\frac{1}{m}\sum_{i=1}^{m} g_i(\boldsymbol{x}))$. We note that the compositional structure in $\Phi(\cdot)$ leads to more challenges in optimization compared with the non-compositional finite-sum problem, since the stochastic gradient of the loss function is not an unbiased estimation of the full gradient. To overcome this issue, and by building on incremental variance reduction (Zhang & Xiao, 2019b), we propose a more general algorithm with a new ingredient in which we also employ variance reduction on the extra term $h(\boldsymbol{x}) = (1/m)\sum_{i=1}^{m} h_i(\boldsymbol{x})$. To handle $r(\cdot)$, similar to Zhang & Xiao (2019b), we assume $r$ is convex and a relatively simple function. We follow the proximal gradient iterates (Beck, 2017; Nesterov, 2013) as follows:

$$\boldsymbol{x}^{(t+1)} = \Pi_r^\eta\left(\boldsymbol{x}^{(t)} - \eta\nabla\Phi(\boldsymbol{x}^{(t)})\right) \tag{8}$$

where we apply the proximal operator of $r(\boldsymbol{x})$ with the learning rate $\eta$ as $\Pi_r^\eta(\boldsymbol{x}) \triangleq \arg\min_{\boldsymbol{y}}\left[r(\boldsymbol{y}) + \frac{1}{2\eta}\|\boldsymbol{y} - \boldsymbol{x}\|^2\right]$. By defining the proximal gradient mapping of $\Psi$ as $\mathcal{G}_\eta(\boldsymbol{x}) \triangleq$

---

**Algorithm 1:** Generalized Composite Incremental Variance Reduction (`GCIVR` ($\boldsymbol{x}^{(0)}$))

---

**Inputs:** Number of iterations $t = 1, \ldots, T$, learning rate $\eta$, initial global model $\boldsymbol{x}^{(0)}$, the size of epoch length $\tau_t$, and mini-batch sizes of $B_t$ and $S_t$ at time $t$.

**for** $t = 1, \ldots, T$ **do**

Sample a mini-batch $\mathcal{B}^{(t)}$ with size $B_t$ uniformly over $[m]$ and compute

$$\boldsymbol{y}_0^{(t)} = \frac{1}{B_t} \sum\nolimits_{\xi \in \mathcal{B}^{(t)}} g(\boldsymbol{x}_{\tau_t}^{(t)}; \xi), \boldsymbol{z}_0^{(t)} = \frac{1}{B_t} \sum\nolimits_{\xi \in \mathcal{B}^{(t)}} \nabla g(\boldsymbol{x}_{\tau_t}^{(t)}; \xi), \boldsymbol{w}_0^{(t)} = \frac{1}{B_t} \sum\nolimits_{\xi \in \mathcal{B}^{(t)}} \nabla h(\boldsymbol{x}_{\tau_t}^{(t)}; \xi)$$

Compute $\tilde{\nabla}\Phi(\boldsymbol{x}_0^{(t)}) = \left(\boldsymbol{z}_0^{(t)}\right)^\top \left(f'(\boldsymbol{y}_0^{(t)})\right) + \boldsymbol{w}_0^{(t)}$

Update the model as follows: $\boldsymbol{x}_1^{(t)} = \Pi_r^\eta \left(\boldsymbol{x}_0^{(t)} - \tilde{\nabla}\Phi(\boldsymbol{x}_0^{(t)}))\right)$

**for** $j = 1, \ldots, \tau_t - 1$ **do in parallel**

Sample a mini-batch $\mathcal{S}_j^{(t)}$ with size $S_t$ uniformly over $[m]$, and form the estimates

$$\boldsymbol{y}_j^{(t)} = \boldsymbol{y}_{j-1}^{(t)} + \frac{1}{S_t} \sum\nolimits_{\xi \in \mathcal{S}_j^{(t)}} \left[g(\boldsymbol{x}_j^{(t)}; \xi) - g(\boldsymbol{x}_{j-1}^{(t)}; \xi)\right] \tag{9}$$

$$\boldsymbol{z}_j^{(t)} = \boldsymbol{z}_{j-1}^{(t)} + \frac{1}{S_t} \sum\nolimits_{\xi \in \mathcal{S}_j^{(t)}} \left[\nabla g(\boldsymbol{x}_j^{(t)}; \xi) - \nabla g(\boldsymbol{x}_{j-1}^{(t)}; \xi)\right] \tag{10}$$

$$\boldsymbol{w}_j^{(t)} = \boldsymbol{w}_{j-1}^{(t)} + \frac{1}{S_t} \sum\nolimits_{\xi \in \mathcal{S}_j^{(t)}} \left[\nabla h(\boldsymbol{x}_j^{(t)}; \xi) - \nabla h(\boldsymbol{x}_{j-1}^{(t)}; \xi)\right] \tag{11}$$

Compute $\tilde{\nabla}\Phi(\boldsymbol{x}_j^{(t)}) = \left(\boldsymbol{z}_j^{(t)}\right)^\top \left(f'(\boldsymbol{y}_j^{(t)})\right) + \boldsymbol{w}_j^{(t)}$

Update the model as follows: $\boldsymbol{x}_{j+1}^{(t)} = \Pi_r^\eta \left(\boldsymbol{x}_j^{(t)} - \tilde{\nabla}\Phi(\boldsymbol{x}_j^{(t)}))\right)$

**end**

**end**

**Output:** Return a randomly selected solution from $\{\boldsymbol{x}_j^t\}_{j=0,\ldots,\tau_t}^{t=1,\ldots,T}$

---

$\frac{1}{\eta}\left[\boldsymbol{x} - \Pi_r^\eta\left(\boldsymbol{x} - \eta\nabla\Phi(\boldsymbol{x})\right)\right]$, the updating rule in Eq. (8) can be equivalently written as $\boldsymbol{x}^{(t+1)} = \boldsymbol{x}^{(t)} - \eta\mathcal{G}_\eta(\boldsymbol{x}^{(t)})$. Given any $\boldsymbol{y}$ as an output of randomized algorithm, we say $\boldsymbol{y}$ achieves the stationary point of problem in Eq. (2) in expectation if $\mathbb{E}\left[\|\mathcal{G}_\eta(\boldsymbol{y})\|^2\right] \leq \epsilon$ holds. Our goal is to achieve an $\epsilon$ stationary point with the least number of calls to a (mini-batch) stochastic oracle.

Focusing on $\Phi(\cdot)$, as detailed in Algorithm 1, we apply three time-scale variance-reduced estimators for $g_i(\boldsymbol{x})$ and its gradient $\nabla g_i(\boldsymbol{x})$, as well as $h_i(\boldsymbol{x})$. For DRO with Wasserstein divergence metric with optimally strongly objective, at the beginning of each epoch $t$ we compute a full-batch gradient over the entire data samples $B_t = m$, i.e., $\boldsymbol{y}_0^{(t)} = \frac{1}{m}\sum_{i=1}^m g_i(\boldsymbol{x}_{\tau_t}^{(t)}), \boldsymbol{z}_0^{(t)} = \frac{1}{m}\sum_{i=1}^m \nabla g_i(\boldsymbol{x}_{\tau_t}^{(t)}), \boldsymbol{w}_0^{(t)} = \frac{1}{m}\sum_{i=1}^m \nabla h_i(\boldsymbol{x}_{\tau_t}^{(t)})$. In contrast, for the case of DRO with $\chi^2$ or KL divergence metrics and non-convex objectives we incrementally increase the size of the mini-batch $B_t \leq m$ at the beginning of each epoch until it reaches the point where we need to compute the full-batch of samples. We should also highlight that a mini-batch in case of Wasserstein DRO indicates a batch of constraints and in DRO with $\chi^2$ or KL divergence metrics represents the number of data sample accessed. We denote the length of each epoch and the mini-batch size within each epoch with $\tau_t$ and $S_t$, respectively. In each epoch $t$ and iteration $j$, we estimate mini-batch gradients from Eqs. (9), (10), and (11) in Algorithm 1, with some corrections term applied from the previous iteration. We note that the variance-reduced term corresponding to the correction is inspired by SARAH (Nguyen et al., 2017) and SPIDER (Fang et al., 2018). Finally, the algorithm returns a randomly selected solution from the iterates.

**Distributed variant of Algorithm 1.** As mentioned before, efficient training of the stochastic composite optimization problem has attracted increasing attention in recent years. Despite much progress, all of existing methods including Algorithm 1 only focus on the single-machine setting. To employ Algorithm 1 in a distributed setting with $p$ machines, in Appendix E we propose a distributed variant of proposed algorithm and establish its convergence rate for convex and non-convex objectives which enjoys a speedup in terms of number of machines.

## 4 CONVERGENCE ANALYSIS

In this section we establish the convergence of proposed algorithm for different DRO notions discussed in Section 2. We start by stating the general assumptions and then discuss the obtained rates. Due to lack of space we only include the rates on the convergence of DRO with Wasserstein metric for strongly convex, and $\chi^2$ and KL divergence metrics for non-convex objectives. We defer the analysis of $\chi^2$ and KL divergence metrics with strongly objectives to Appendix D. To establish the convergence rates, we first introduce some standard assumptions.

**Assumption 1.** *We make the following assumptions on the components of objective* $\Psi(\boldsymbol{x}) = r(\boldsymbol{x}) + \frac{1}{m}\sum_{i=1}^m h_i(\boldsymbol{x}) + f(\frac{1}{m}\sum_{i=1}^m g_i(\boldsymbol{x}))$:

1) $\|\nabla h(\boldsymbol{x}_1) - \nabla h(\boldsymbol{x}_2)\| \leq L_h \|\boldsymbol{x}_1 - \boldsymbol{x}_2\|$ *where* $\boldsymbol{x}_1, \boldsymbol{x}_2 \in \mathbb{R}^d$. *We also assume that* $\|h(\boldsymbol{x}_1) - h(\boldsymbol{x}_2)\| \leq \ell_h \|\boldsymbol{x}_1 - \boldsymbol{x}_2\|$.
2) $\|f'(\boldsymbol{x}_1) - f'(\boldsymbol{x}_2)\| \leq L_f \|\boldsymbol{x}_1 - \boldsymbol{x}_2\|$ *where* $\boldsymbol{x}_1, \boldsymbol{x}_2 \in \mathbb{R}$. *We also assume that* $\|f(\boldsymbol{x}_1) - f(\boldsymbol{x}_2)\| \leq \ell_f \|\boldsymbol{x}_1 - \boldsymbol{x}_2\|$.
3) $\|\nabla g(\boldsymbol{x}_1; \xi) - \nabla g(\boldsymbol{x}_2; \xi)\| \leq L_g \|\boldsymbol{x}_1 - \boldsymbol{x}_2\|$ *and* $\|g(\boldsymbol{x}_1; \xi) - g(\boldsymbol{x}_2; \xi)\| \leq \ell_g \|\boldsymbol{x}_1 - \boldsymbol{x}_2\|$ *for* $\forall \boldsymbol{x}_1, \boldsymbol{x}_2 \in \mathbb{R}^d$.
4) *We suppose that* $\Psi(\boldsymbol{x})$ *in Eq. (2) is bounded from below that* $\Psi^* = \inf_{\boldsymbol{x}} \Psi(\boldsymbol{x}) > -\infty$.
5) *We assume* $r(\boldsymbol{x}) \in \mathbb{R} \cup \{\infty\}$ *is a convex and lower-semicontinues function.*

An immediate implication of above assumption is that $f(g(\boldsymbol{x})) \triangleq f(\frac{1}{m}\sum_{i=1}^m g_i(\boldsymbol{x}))$ is smooth with module $(\ell_g^2 L_f + \ell_f L_g)$, hence $\Psi(\boldsymbol{x})$ is smooth with module $L_\Phi \triangleq \left[(\ell_g^2 L_f + \ell_f L_g) + L_h\right]$ (Wang et al., 2016; Zhang & Xiao, 2019b).

**Assumption 2** (Unbiased estimation). *We assume that the mini-batch of samples* $\xi$ *over functions* $g_i$ *for* $i = 1, 2, \ldots, m$ *is unbiased, that is* $\mathbb{E}[g(\boldsymbol{x}; \xi)] = g(\boldsymbol{x})$ *and* $\mathbb{E}[\nabla g(\boldsymbol{x}; \xi)] = \nabla g(\boldsymbol{x})$.

### 4.1 OPTIMALLY STRONGLY CONVEX OBJECTIVES

We here establish the convergence for optimally strongly convex objectives under Wasserstein metric.

**Definition 1** (Optimally strongly convex). *We say that the objective function* $\Psi(\boldsymbol{x})$ *is optimally strongly convex objective with module* $\mu$ *if* $\Psi(\boldsymbol{x}) - \Psi(\mathcal{P}_{\boldsymbol{x}^*}(\boldsymbol{x})) \geq \mu \|\boldsymbol{x} - \mathcal{P}_{\boldsymbol{x}^*}(\boldsymbol{x})\|^2$.

According to Definition 4 of Necoara et al. (2019) optimally strong objectives or quadratic functional growth property is the generalization of strongly convex condition. According to Karimi et al. (2016) below, the PL condition implies an optimally strongly convex condition but not vice versa. Therefore, optimally strongly convex generalizes PL condition as well.

Following Cotter et al. (2016) and Mahdavi et al. (2012), we make the following assumption.

**Assumption 3.** *There exists a constant* $\rho$ *such that if we define* $g(\boldsymbol{x}) = \max_{1 \leq i \leq m} \tilde{g}_i(\boldsymbol{x})$ *we have* $\min_{g(\boldsymbol{x})=0} \|\nabla g(\boldsymbol{x})\|_2 \geq \rho$.

**Assumption 4.** *Function* $r(\boldsymbol{x})$ *is smooth with module* $G$.

**Theorem 4.1.** *Assume* $\Psi$ *is* $\mu$-*optimally strongly convex and set* $\tau_t = S_t = \sqrt{B_t} = \sqrt{m}$, $T = \frac{5}{\sqrt{m}\mu\eta}$, $\eta < \frac{2}{L_\Phi + \sqrt{L_\Phi^2 + 36 G_0}}$ *where* $G_0 \triangleq 3(\ell_g^4 L_f^2 + \ell_f^2 L_g^2 + \ell_h^2)$, $\alpha > \frac{G}{\rho}$, *and* $\gamma = \frac{\exp(-K)}{\ln(m+1)}$. *Let us denote* $\boldsymbol{x}^{(k+1)} = \texttt{GCIVR}(\boldsymbol{x}^{(k)})$ *for* $k = 0, \ldots, K-1$ *(using Algorithm 1). Under Assumptions 1-4 and by letting* $K = \ln(1/\epsilon)$, *the solution of DRO with Wasserstein divergence is obtained by projecting* $\boldsymbol{x}^{(K)}$ *onto the constraint set* $\mathcal{K} = \{\boldsymbol{x} | g_i(\boldsymbol{x}) \leq 0, i = 1, 2, \ldots, m\}$, *i.e.,* $\bar{\boldsymbol{x}}^{(K)} = \Pi_{\mathcal{K}}(\boldsymbol{x}^{(K)})$. *In order to achieve* $r(\bar{\boldsymbol{x}}^{(K)}) - r(\boldsymbol{x}^{(*)}) \leq \epsilon$, *we require an* $O\left((m + \kappa\sqrt{m})\ln\frac{1}{\epsilon}\right)$ *calls to the stochastic oracle.*

**Comparison with previous results.** Compared to the MidTouch approach by Cotter et al. (2016), our obtained rate improves both on the dependency on the number of constraints $m$, as well as convergence error $\epsilon$. To better understand the intuition behind achieving such a double folded improvement in terms of $m$ and $\epsilon$, we note that unlike Cotter et al. (2016) which utilizes a primal-dual approach in order to obtain an approximate to the optimal distribution over the constraints, we directly find the optimal distribution exactly. Second, while Cotter et al. (2016) does not apply any variance reduction technique to primal variable $\boldsymbol{x}$, our algorithm benefits from variance reduction over $\boldsymbol{x}$ too.

Furthermore, Theorem 4.1 entails tighter rate in terms of final accuracy compared to Yu et al. (2021). These results are summarized in Table. 1.

In Algorithm 1 for optimally strongly convex objectives, we need to do a single projection at the end. The following corollary bounds the error between the solution before and after the projection.

**Corollary 4.2.** *Under the assumptions made in Theorem 4.1, the error between the solution with and without projection is bounded by*

$$r(\bar{\boldsymbol{x}}^{(K)}) - r(\boldsymbol{x}^{(K)}) \leq G \left[ \frac{\gamma \ln(m+1)}{\alpha\rho - G} + \frac{1}{\alpha\rho - G} O\left(\exp(-K)\right) \right] + \gamma \ln(m+1). \tag{12}$$

*Therefore, with a proper choice of $\gamma$ we can establish convergence rate obtained in Theorem 4.1.*

**Remark 1.** *It is worthy to highlight that through reduction of heavily-constrained optimization to composite optimization,* **GCIVR** *algorithm can be considered as an alternative method to solving constrained optimization problems via mini-batch sampling of constraints. In particular, Theorem 4.1 shows that under certain conditions, we can solve any heavily-constrained optimization problem with sampling a mini-batch of constraints and achieve similar guarantees compared to projection-based counterparts while avoiding the heavy dependency on the number of constraints. Another implication of Theorem 4.1 is that we can solve heavily constrained optimization with the sample complexity similar to that of an unconstrained optimization problem.*

**Remark 2.** *To understand the tightness of our obtained rate, consider* $\min_{\boldsymbol{x}} \left[ \Psi(\boldsymbol{x}) \triangleq r(\boldsymbol{x}) + \frac{1}{m}\sum_{i=1}^{m} g_i(\boldsymbol{x}) \right]$ *which is an instance of our general optimization problem (2). Clearly, any lower bound to solve this instance also holds for the original optimization problem (2). Xie et al. (2019) provides a lower-bound of $O\left((m + \sqrt{\kappa m}) \ln(1/\epsilon)\right)$ for above instance, matching our upper bound in terms of the dependency on the number of data samples, while the dependency on $\kappa$ can still be improved. Furthermore, for the DRO problem with Wasserstein divergence metric, rather than solving constrained optimization problem we solve unconstrained compositional optimization problem. Thus, since solving constrained optimization is more complex than unconstrained optimization problem, we do not expect to obtain any better bound regarding $m$ even for DRO with Wasserstein metric.*

**Remark 3.** *In a distributed setting, we are able to improve the sample complexity to* $O\left(\left(m + \frac{m}{p} + \kappa\sqrt{m}\right)\ln\frac{1}{\epsilon}\right)$ *with $p$ devices. The proof can be found in Appendix F.3.*

## 4.2 NON-CONVEX OBJECTIVES

We now turn to establishing the convergence of DRO problems with $\chi^2$ and KL divergence metrics for non-convex objectives by making an additional assumption on the variance of stochastic mini-batches.

**Assumption 5** (Bounded variance). *The mini-batch sampling has bounded variance that is* $\mathbb{E}\left[\|\nabla g(\boldsymbol{x};\xi) - \nabla g(\boldsymbol{x})\|^2\right] \leq \frac{\sigma^2}{B}$, *where $B$ indicates the batch size.*

**Theorem 4.3.** *Under Assumptions 1,2, and 5 using Algorithm 1 for some positive constants $\beta$ and $0 \leq \zeta < \sqrt{m}$, denote $T_0 \triangleq \frac{\sqrt{m}-\zeta}{\beta} = O\left(\sqrt{m}\right)$. For $t \leq T_0$, let us set $\tau_t = S_t = \sqrt{B_t} = \beta t + \zeta$ and for $t > T_0$ set $\tau_t = S_t = \sqrt{m}$. Then, if $\eta < \frac{4}{L_\Phi + \sqrt{L_\Phi^2 + 12G_0}}$, after $T = \tilde{O}\left(\min(1/\sqrt{\epsilon}, 1/\sqrt{m}\epsilon)\right)$ iterations, with $\tilde{O}\left(\min\{\sqrt{m}/\epsilon, 1/\epsilon^{1.5}\}\right)$ number of calls to the stochastic oracle it holds that $\mathbb{E}\left[\left\|\mathcal{G}_\eta(\boldsymbol{x}^{(T)})\right\|^2\right] \leq \epsilon$, where we used $\tilde{O}(.)$ notation to hide terms with logarithmic dependency.*

We also remark that Assumption 5 is only needed for convergence until $T_0 = \frac{\sqrt{m}-\zeta}{\beta}$ and after $t > T_0$ we will not need this assumption. We also note that this assumption is not required in optimally strongly convex setting as we utilize a full batch at the beginning of each epoch.

**Discussion on lower bound.** As we discussed in Remark 2, any impossibility result for optimizing $\min_{\boldsymbol{x}} \left[ \Psi(\boldsymbol{x}) \triangleq r(\boldsymbol{x}) + \frac{1}{m}\sum_{i=1}^{m} g_i(\boldsymbol{x}) \right]$ also holds for general compositional optimization problem in Eq. (2). For former problem, Fang et al. (2018) shows that to achieve the stationary point with error $\epsilon$ we require at least $\tilde{O}\left(\min\{\frac{\sqrt{m}}{\epsilon}, \frac{1}{\epsilon^{1.5}}\}\right)$ (almost optimal) gradient oracle calls for strongly optimal objectives. As a consequence, for the DRO with $\chi^2$ or KL divergence metrics we do not expect less number of constraint checks or sample complexity, respectively.

**Remark 4.** *In distributed setting with $p$ devices, we can improve the convergence rate of DRO with $\chi^2$ and KL metrics to $\tilde{O}\left(\min\{\frac{\sqrt{m}}{p\epsilon} + \frac{\sqrt{m}}{\epsilon}, \frac{1}{p\epsilon^{1.5}} + \frac{1}{\epsilon^{1.5}}\}\right)$. For details, please refer to Appendix F.3.*

## 5 EXPERIMENTS

In this section, we empirically examine the efficacy of the proposed algorithm in different use cases. The main algorithm that we compare against is Heavily-Constrained algorithm that uses a parametric multiplier model proposed in Narasimhan et al. (2020b). We call this algorithm Heavily-constrained, where they learn the Lagrange multiplier values using a parametric model such as a neural network. The tasks we apply the algorithms are mainly focused on fairness constraints since they fit greatly to the problem definition in this paper due to the large number of constraints they add to the main learning problem. In addition, similar to distribution robust optimization approaches, the main goal of fairness constraint is to find a solution that is agnostic to the distribution of protected groups. The code for the experiments is available at this repository.

**Distributionally robust optimization for fairness constraints.** The first experiment is based on Narasimhan et al. (2020b); Wang et al. (2020), where we want to enforce equality of opportunity (Hardt et al., 2016) constraints for different groups while the group membership is noisy and changing during the training. Hence, the problem is to make the solution distributionally robust among different protected groups in the problem. Based on the setup in Narasimhan et al. (2020b), we assume we have access to the marginal probability of the true groups ($\mathbb{P}(g_i = j|\hat{g}_i = k)$, where $g_i$ is the true group membership and $\hat{g}_i$ is the noisy group membership). Hence, to enforce fairness constraints, we consider all possible proxy groups using this marginal probability, which can increase the number of constraints greatly. The goal in this case for equal opportunity is to have the true positive rate of each group in the vicinity of the true positive rate (tpr) of the overall data, that is: $\mathsf{tpr}(g = j) \geq \mathsf{tpr}(ALL) - \epsilon$ for every proxy group we define.

In this experiment, we use the Adult dataset (Dua & Graff, 2017), and consider race groups of "white", "black", and "other" as protected groups. We train a linear classifier with logistic regression, and report the overall error rate of the classifier, as well as the maximum violation of the fairness constraints (equal opportunity) over true group memberships. We set $\epsilon = 0.05$ and the noise level to 0.3. Again, we compare with unconstrained optimization and heavily-constrained algorithm with a linear model as its multiplier model. First row of Figure 1 shows the results for this experiment, where the proposed **GCIVR** algorithm can achieve the same level of constraint violation on true group memberships as the heavily constraint while outperforms it in terms of overall error rate. Hence, the solution found by the **GCIVR** dominates heavily-constrained solution. In terms of runtime speed, the first row of Figure 1(c) clearly shows the advantage of **GCIVR** over heavily-constrained with much lower overhead to the unconstrained optimization.

**Fairness Constraints on intersectional groups.** In this task we ought to learn a linear classifier with logistic regression to predict the crime rate for a community on Communities and Crime dataset (Dua & Graff, 2017). This dataset contains 1994 instances of communities each with 128 features to predict the per capita crime rate for each community in the US. The labels are high and low crime rates to represent if a community is above the 70th percentile of the data or not. To add fairness constraints, we first determine different communities based on the percentages of the Black, Hispanic and Asian population in each community, as discussed in Cotter et al. (2019); Narasimhan et al. (2020b). Similar to Narasimhan et al. (2020b), we generate 1000 thresholds of the form $(\tau_1, \tau_2, \tau_3) \in [0, 1]^3$ to define each group. Consider the population of each race as $(p_1, p_2, p_3)$, then a community belongs to group $g_{\tau_1, \tau_2, \tau_3}$ if $p_i \geq \tau_i \,\forall i \in [3]$. Then the fairness constraints enforce that the error rate of each group should be in the neighborhood of the overall error by margin of $\epsilon$. That is $\mathsf{err}(g_{\tau_1, \tau_2, \tau_3}) \leq \mathsf{err}(ALL) + \epsilon$. Similar to Narasimhan et al. (2020b), we set $\epsilon = 0.01$ and not consider groups with less than $1\%$ of data.

We compare with the unconstrained optimization and Heavily-constrained algorithm with a 2-layer neural network, each with 100 nodes as their Lagrange model, as described in their paper. We compare the trade-off between the error rate and maximum violation of fairness constraints among groups. The second row of Figure 1 portrays this comparison for the test dataset. As it can be inferred, the proposed **GCIVR** algorithm achieves the same error rate as Heavily-constrained, but both higher than the unconstrained optimization. However, comparing the maximum violation of constraints, it is clear that **GCIVR** outperforms both optimization methods. Also, the third figure shows the runtime of different algorithms. It is clear that **GCIVR** adding a minimal overhead to the unconstrained optimization, while heavily-constrained increases the runtime more than 100 times.

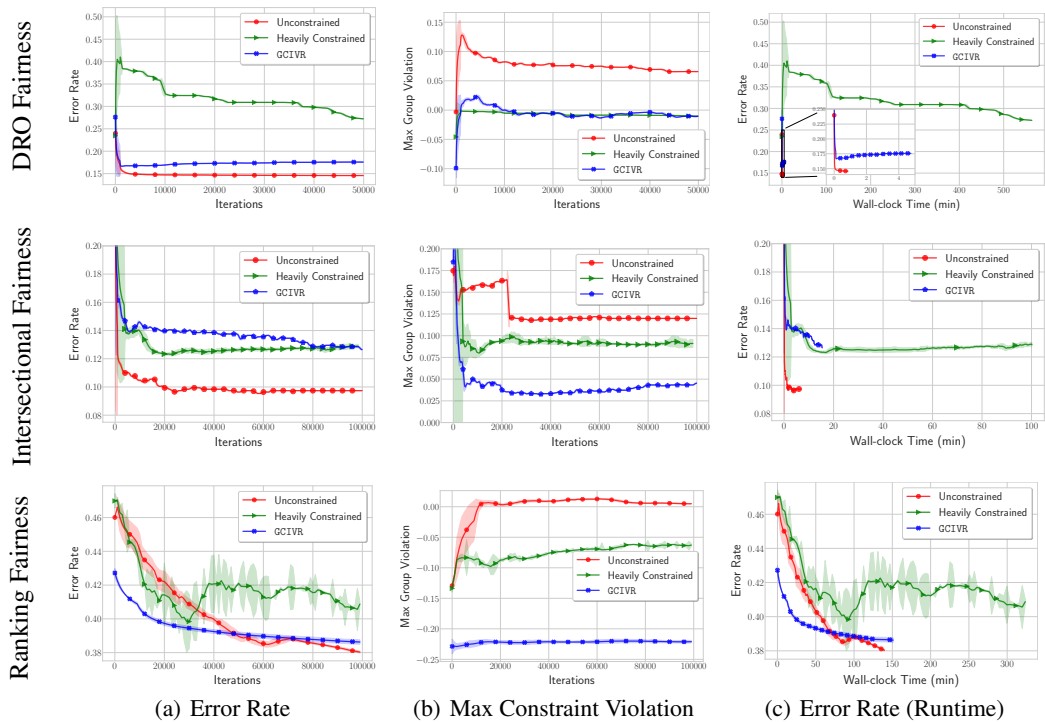

(a) Error Rate     (b) Max Constraint Violation     (c) Error Rate (Runtime)

Figure 1: Comparison of the proposed `GCIVR` algorithm with unconstrained optimization and heavily-constrained algorithm Narasimhan et al. (2020b) in three different tasks of fairness and DRO. Each row shows the result for one task based on the error rate, max constraint violations and the runtime of the codes. In all the cases solutions learned by `GCIVR` dominates the heavily-constrained solution and it converges way faster.

**Per-query fairness constraints in ranking.** In the third problem we evaluate our algorithm on the fairness in the ranking problem, where we intend to impose per-query constraints on the learning to rank problem as defined in Narasimhan et al. (2020a;b). We divide document-query pairs into two groups based on the 40th percentile of the their QualityScore features. Hence, in this problem, we want to learn a ranking function $f : \mathcal{D} \times \mathcal{Q} \to \mathbb{R}$, which maps a pair of document-query features to a real number as the score. Consider the groups of $g_0$ and $g_1$ as mentioned before, we want the error rate for these groups to be close to each other. In other terms, if the pairwise error rate is defined as: $\mathsf{err}_{i,j}(q) = \mathbb{E}\left[\mathbf{1}\left\{f(d,q) < f\left(d',q\right)\right\} \mid y > y', d \in g_i, d' \in g_j\right]$, where $y$ and $y'$ are the respective binary labels. Then, the fairness constraints can be satisfied as $|\mathsf{err}_{0,1}(q) - \mathsf{err}_{1,0}(q)| \leq \epsilon \ \forall q \in \mathcal{Q}$.

For this experiment, we use Microsoft Learning to Rank Dataset (MSLR-WEB10K) (Qin & Liu, 2013), which contains 10K queries and 136 features. For this experiment we use a non-convex objective, where the model is a two-layer neural network each with 128 nodes and cross-entropy as the loss function. We compare against the unconstrained optimization and the heavily-constrained algorithm with an one-layer neural network with 64 nodes as its multiplier model. We use 1000 queries in the training and 100 queries in the test datasets. The third row in Figure 1 shows the error rate and maximum violation of groups constraints. As it is clear `GCIVR` outperforms heavily-constrained in both error rate and maximum violations of groups by a large margin. In terms of runtime, `GCIVR` is very close to the unconstrained optimization, and about $2\times$ faster than heavily-constrained.

## 6 CONCLUSION

In this paper, we showed that many DRO problems or heavily constrained optimization problems can be cast into a general framework based on finite-sum composite optimization. To solve this composite finite-sum optimization, we introduced centralized and distributed algorithms. We theoretically illustrated that our algorithm converges with an almost optimal number of constraint checks (for Wasserstein distance) or gradient calls (for $\chi^2$ or KL divergence metrics). Finally, we validated our theory with a number of experiments.

## ACKNOWLEDGEMENTS

Amin Karbasi acknowledges funding in direct support of this work from NSF (IIS-1845032) and ONR (N00014-19-1-2406). The work of Mehrdad Mahdavi was supported in part by NSF (CNS-1956276).

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

# Appendix

In the appendix, we provide the missing proofs and derivations from the main manuscript, as well as proposing the distributed version of Algorithm 1 to further improve the convergence speed.

## TABLE OF CONTENTS

## A  EXAMPLE OF EQUIVALENT WESSTERSTEIN REFORMULATION

A tractable reformulation of distributionally robust logistic regression problem by reduction to a constrained optimization problem is provided in Shafieezadeh-Abadeh et al. (2015) as follows:

$$\min_{\boldsymbol{x}} \qquad r(\boldsymbol{x}) \triangleq \lambda\epsilon + \frac{1}{m}\sum_{i=1}^{m} s_i$$

$$\text{subject to} \quad g_i(\boldsymbol{x}) = (\ell_{\boldsymbol{\beta}}(\boldsymbol{z}, y) - s_i) \leq 0, \ \forall i \in [1:m]$$

$$g_j(\boldsymbol{x}) = (\ell_{\boldsymbol{\beta}}(\boldsymbol{z}, -y) - \lambda\kappa - s_j) \leq 0, \ \forall j \in [1:m]$$

$$g_\ell(\boldsymbol{x}) = \left(\|\boldsymbol{\beta}\|_*^2 - \lambda^2\right) \leq 0 \tag{13}$$

where $\ell_{\boldsymbol{\beta}}(\boldsymbol{z}, y) = \log\left(1 + \exp\left(-y\langle\boldsymbol{\beta}, \boldsymbol{z}\rangle\right)\right)$ is the associate loss with parameter $\boldsymbol{\beta}$ and the data sample $(\boldsymbol{z}, y)$ and we define $\boldsymbol{x} = (\boldsymbol{z}, s_1, \ldots, s_m, \lambda, \boldsymbol{\beta})$.

## B  OBTAINING AUGMENTED OPTIMIZATION PROBLEM

To derive the smoothed constrained variant of original optimization problem we follow the reduction method originally introduced in Mahdavi et al. (2012) to get:

$$\min_{\boldsymbol{x}} \bar{\Psi}(\boldsymbol{x}) = \min_{\boldsymbol{x}}\left[r(\boldsymbol{x}) + \max_{0 \leq p_i \leq \alpha, \ \sum_{i=1}^{m+1} p_i = \alpha}\left[\sum_{i=1}^{m} p_i \tilde{g}_i(\boldsymbol{x}) + \gamma H(\frac{p_1}{\alpha}, \frac{p_2}{\alpha}, \ldots, \frac{p_m}{\alpha}, \frac{p_{m+1}}{\alpha}, \frac{1}{m+1})\right]\right]$$

$$= \min_{\boldsymbol{x}}\left[r(\boldsymbol{x}) + \gamma\ln\left(1 + \sum_{i=1}^{m}\exp\left(\frac{\alpha\tilde{g}_i(\boldsymbol{x})}{\gamma}\right)\right)\right]$$

$$= \min_{\boldsymbol{x}}\left[r(\boldsymbol{x}) + \gamma\ln\left(\frac{1}{m+1}\left[1 + \sum_{i=1}^{m}\exp\left(\frac{\alpha\tilde{g}_i(\boldsymbol{x})}{\gamma}\right)\right]\right) + \gamma\ln(m+1)\right]$$

$$= \min_{\boldsymbol{x}}\left[\Psi(\boldsymbol{x}) + \underbrace{\gamma\ln(m+1)}_{\text{Constant}}\right], \tag{14}$$

where in Eq. (14) we used the definitions $\Psi(\boldsymbol{x}) = r(\boldsymbol{x}) + \gamma\ln(g(\boldsymbol{x}))$, $g_i(\boldsymbol{x}) \triangleq \exp\left(\frac{\alpha\tilde{g}_i(\boldsymbol{x})}{\gamma}\right)$ and $g(\boldsymbol{x}) = \frac{1}{m+1}\left[1 + \sum_{i=1}^{m} g_i(\boldsymbol{x})\right]$. Since $\gamma\ln(m+1)$ is a constant, the minimizer of both $\bar{\Psi}(\boldsymbol{x})$ and $\Psi(\boldsymbol{x})$ is the same, and consequently we can solve the optimization problem for $\Psi$ via finite-sum composite optimization.

## C  PROOF OF CLAIM 2.1

We use Lagrangian multiplier for the purpose of the proof. The Lagrangian of the optimization problem in Eq. (4) is

$$L(\boldsymbol{x}, \gamma; \lambda) = \max_{0 \leq p_i \leq 1, \ \sum_{i=1}^{m} p_i = 1}\left[\sum_{i=1}^{m} p_i f_i(\boldsymbol{x}_i) - \gamma\frac{1}{2m}\sum_{i=1}^{m}(mp_i - 1)^2 - \lambda\left(\sum_{i=1}^{m} p_i - 1\right)\right] \tag{15}$$

By setting $\nabla_{p_i} L = 0$ we obtain

$$p_i = \frac{1}{m}\left(\frac{f_i(\boldsymbol{x}) - \lambda}{\gamma} + 1\right). \tag{16}$$

Then from the condition $\sum_i p_i = 1$, it follows that:

$$\lambda = \frac{1}{m}\sum_{i=1}^{m} f_i(\boldsymbol{x}) \tag{17}$$

Plugging the obtained values for $\lambda$ and $p_i$ into Eq. (4) yields

$$
\begin{aligned}
\Psi(\boldsymbol{x};\gamma) &= 1 + \sum_{i=1}^{m} \frac{1}{\gamma m} \left[ \left( f_i(\boldsymbol{x}) - \frac{1}{m}\sum_{i=1}^{m} f_i(\boldsymbol{x}) \right) - \frac{1}{2} \left( f_i - \frac{1}{m}\sum_{i=1}^{m} f_i(\boldsymbol{x}) \right)^2 \right] \\
&= 1 - \frac{1}{2\gamma \times m} \sum_{i=1}^{m} \left( f_i(\boldsymbol{x}) - \frac{1}{m}\sum_{i=1}^{m} f_i(\boldsymbol{x}) \right)^2 \\
&= 1 - \frac{1}{2\gamma m} \sum_{i=1}^{m} [f_i(\boldsymbol{x})]^2 + \frac{1}{2\gamma} \left[ \frac{1}{m}\sum_{i=1}^{m} f_i(\boldsymbol{x}) \right]^2
\end{aligned}
\tag{18}
$$

which completes the proof.

## D  STRONGLY-CONVEX CONVERGENCE FOR DRO WITH KL AND $\chi^2$ METRICS

**Theorem D.1.** *Suppose Assumptions 1 and 2 hold, and also assume $\Psi$ is $\mu$-optimally strongly convex. If we set $\tau_t = S_t = B_t = \sqrt{m}$ and $T = \frac{5}{\sqrt{m}\mu\eta}$, letting $\eta < \frac{2}{L_\Phi + \sqrt{L_\Phi^2 + 36G_0}}$, by letring $\boldsymbol{x}^{(k+1)} = \textbf{GCIVR}\left(\boldsymbol{x}^{(k)}\right)$ for $k = 0, \ldots, K-1$ using Algorithm 1 after $K = \ln(1/\epsilon)$ repetition solves the optimization problem in Eq. (2) with convergence error $\Psi(\bar{\boldsymbol{x}}^{(T)}) - \Psi(\boldsymbol{x}^{(*)}) \leq \epsilon$ with $O\left((m + \kappa\sqrt{m})\ln\frac{1}{\epsilon}\right)$ number of gradient calls. Therefore, depending on the objective function $\Psi$, this sample complexity corresponds to both DRO problem with $\chi^2$ and regularized KL metrics.*

Based on Theorem D.1, for DRO problem with $\chi^2$ divergence metric, we can achieve $\Psi(\bar{\boldsymbol{x}}^{(K)})|_{\gamma=1} - \Psi(\boldsymbol{x}^{(*)})|_{\gamma=1} \leq \gamma\epsilon$ and $\Psi(\bar{\boldsymbol{x}}^{(K)})|_{\gamma>1} - \Psi(\boldsymbol{x}^{(*)})|_{\gamma>1} \leq \epsilon$ with $O\left((m + \kappa\sqrt{m})\ln\frac{1}{\epsilon}\right)$ number of gradient oracle calls.

## E  DISTRIBUTED STOCHASTIC COMPOSITE ALGORITHM

In the main body, we mainly focused on studying the sample complexity of solving DRO problems in a centralized setting. The question we are interested is that "Can we further improve the sample complexity?" In this section, we give an affirmative answer to this question via studying distributed version of DRO problem. We suppose data samples are distributed among $p$ clients [1]. For distributionally robust optimization problem via Wasserstein , $\chi^2$ and KL divergence metrics in the distributed fashion like federated learning setups McMahan et al. (2016); Mohri et al. (2019); Haddadpour et al. (2019b); Haddadpour & Mahdavi (2019); Haddadpour et al. (2019a); Deng et al. (2020); Haddadpour et al. (2020).

---

[1]We suppose that each device has access to at least $m/p$ number of data points.

We first introduce our distributed finite-sum compositional algorithm as detailed in Algorithm 2.

---

**Algorithm 2:** Distributed Generalized Composite Incremental Variance Reduction (`DistGCIVR`($\boldsymbol{x}^{(0)}$))

---

**Inputs:** Number of iterations $t = 1, \ldots, T$, learning rate $\eta$, initial global model $\boldsymbol{x}^{(0)}$, and a set of triples $\{\tau_t, B_t, S_t\}$ where $\tau_t$ indicating the size of epoch length and mini-batch sizes of $B_t$ and $S_t$ at time $t$.

**for** $t = 1, \ldots, T$ **do**

    **for** $i = 1, \ldots, p$ **do in parallel**

        Sample minibatches $\mathcal{B}^{(t,i)}$ with size $B_t$ uniformly over $[m]$ and compute

$$\boldsymbol{y}_0^{(t,i)} = \frac{1}{B_t} \sum_{\xi \in \mathcal{B}^{(t,i)}} g_i(\boldsymbol{x}_{\tau_t}^{(t)}; \xi), \quad \boldsymbol{z}_0^{(t,i)} = \frac{1}{B_t} \sum_{\xi \in \mathcal{B}^{(t,i)}} \nabla g_i(\boldsymbol{x}_{\tau_t}^{(t)}; \xi), \quad \boldsymbol{w}_0^{(t,i)} = \frac{1}{B_t} \sum_{\xi \in \mathcal{B}^{(t,i)}} \nabla h_i(\boldsymbol{x}_{\tau_t}^{(t)}; \xi)$$

        Server computes $\boldsymbol{y}_0^{(t)} = \frac{1}{p} \sum_{i=1}^p \boldsymbol{y}_0^{(t,i)}$, $\boldsymbol{z}_0^{(t)} = \frac{1}{p} \sum_{i=1}^p \boldsymbol{z}_0^{(t,i)}$ and $\boldsymbol{w}_0^{(t)} = \frac{1}{p} \sum_{i=1}^p \boldsymbol{w}_0^{(t,i)}$

        Server computes $\tilde{\nabla}\Phi(\boldsymbol{x}_0^{(t)}) = \left(\boldsymbol{z}_0^{(t)}\right)^T \left(f'(\boldsymbol{y}_0^{(t)})\right) + \boldsymbol{w}_0^{(t)}$

        Update the model as follows and broadcasts it to devices

$$\boldsymbol{x}_1^{(t)} = \Pi_r^\eta \left(\boldsymbol{x}_0^{(t)} - \tilde{\nabla}\Phi(\boldsymbol{x}_0^{(t)}))\right)$$

        **for** $j = 1, \ldots, \tau_t - 1$ **do in parallel**

            Sample a set $\mathcal{S}_j^{(t,i)}$ with size $S_t$ over $[m]$, and form the estimates

$$\boldsymbol{y}_j^{(t,i)} = \boldsymbol{y}_{j-1}^{(t,i)} + \frac{1}{S_t} \sum_{\xi \in \mathcal{S}_j^{(t,i)}} \left[g_i(\boldsymbol{x}_j^{(t)}; \xi) - g_i(\boldsymbol{x}_{j-1}^{(t)}; \xi)\right]$$

$$\boldsymbol{z}_j^{(t,i)} = \boldsymbol{z}_{j-1}^{(t,i)} + \frac{1}{S_t} \sum_{\xi \in \mathcal{S}_j^{(t,i)}} \left[\nabla g_i(\boldsymbol{x}_j^{(t)}; \xi) - \nabla g_i(\boldsymbol{x}_{j-1}^{(t)}; \xi)\right] \tag{19}$$

$$\boldsymbol{w}_j^{(t,i)} = \boldsymbol{w}_{j-1}^{(t,i)} + \frac{1}{S_t} \sum_{\xi \in \mathcal{S}_j^{(t,i)}} \left[\nabla h_i(\boldsymbol{x}_j^{(t)}; \xi) - \nabla h_i(\boldsymbol{x}_{j-1}^{(t)}; \xi)\right] \tag{20}$$

            and send $\boldsymbol{y}^{(t,i)}$ and $\boldsymbol{z}^{(t,i)}$ back to server

            Server computes $\boldsymbol{y}_j^{(t)} = \frac{1}{p} \sum_{i=1}^p \boldsymbol{y}_j^{(t,i)}$, $\boldsymbol{z}_j^{(t)} = \frac{1}{p} \sum_{i=1}^p \boldsymbol{z}_j^{(t,i)}$ and

            $\boldsymbol{w}_j^{(t)} = \frac{1}{p} \sum_{i=1}^p \boldsymbol{w}_j^{(t,i)}$

            Compute $\tilde{\nabla}\Phi(\boldsymbol{x}_j^{(t)}) = \left(\boldsymbol{z}_j^{(t)}\right)^T \left(f'(\boldsymbol{y}_j^{(t)})\right) + \boldsymbol{w}_j^{(t)}$

            Update the model as follows: $\boldsymbol{x}_{j+1}^{(t)} = \Pi_r^\eta \left(\boldsymbol{x}_j^{(t)} - \tilde{\nabla}\Phi(\boldsymbol{x}_j^{(t)}))\right)$

        **end**

    **end**

**end**

**Output:** Return a randomly selected solution from $\{\boldsymbol{x}_j^t\}_{j=0,\ldots,\tau_t}^{t=1,\ldots,T}$

---

We emphasize that our algorithm is developed based on Algorithm 1, so we only describe the main differences. In the the distributed setting, in epoch $t$, $i$-th device has local version of global model, i.e., $\boldsymbol{y}_j^{(t,i)}, \boldsymbol{z}_j^{(t,i)}, \boldsymbol{w}_j^{(t,i)}$ and at the beginning and during the epochs all devices send back their local models back to the server to be averaged and global model $\boldsymbol{x}$ to be updated. Then, server broadcast global model to the devices.

## F  PROOF OF THEOREMS 4.1 AND F.2

Before proceeding to the proof of this theorem, we need to mention that for Wasserstein DRO problem the main objective $r(\boldsymbol{x})$ is an affine function of input data. This property naturally satisfies condition (5) in Assumption 1 and will be useful in distributed setting. For an illustrative example please see Appendix A.

**Proof of Theorem 4.1:** Suppose that $\boldsymbol{x}^{(*)}$ is the solution for the following optimization problem:

$$\underset{\boldsymbol{x}}{\text{minimize}} \quad r(\boldsymbol{x}) \triangleq \frac{1}{m} \sum_{i=1}^{m} f_i(\boldsymbol{x}_i) \tag{21}$$

$$\text{subject to} \quad \tilde{g}_i(\boldsymbol{x}) \leq 0, \ \forall i \in \{1, \cdots, m\}.$$

Defining $\Psi(\boldsymbol{x}) \triangleq r(\boldsymbol{x}) + \gamma \ln\left(\frac{1}{m+1}\left[1 + \sum_{i=1}^{m} \exp\left(\frac{\alpha \tilde{g}_i(\boldsymbol{x})}{\gamma}\right)\right]\right)$, we have

$$\bar{\Psi}(\boldsymbol{x}) \triangleq \Psi(\boldsymbol{x}) + \gamma \ln(m+1).$$

Noting that $\bar{\boldsymbol{x}}^{(K)}$ and $\boldsymbol{x}^{(K)}$ are the output of algorithm with and without projection to the constraints set

$$\mathcal{K} = \{\boldsymbol{x} : \tilde{g}_i(\boldsymbol{x}) \leq 0, \forall i \in [1:m]\} \tag{22}$$

respectively. Our proof is based on following two steps:

(1) We first show that $\mathbb{E}\left[\Psi(\boldsymbol{x}^{(K)})\right] \leq \Psi(\boldsymbol{x}^{(*)}) + O\left(\exp(-aK)\right)$, which indicates the convergence rate of stochastic compositional optimization problem depends on used algorithm without any projection.

(2) Second step involves showing that $\mathbb{E}\left[r(\bar{\boldsymbol{x}}^{(K)})\right] \leq r(\boldsymbol{x}^{(*)}) + O\left(\exp(-aK)\right)$; in other words the final projected solution converges to optimal solution of augmented objective function and $a$ is some positive constant depends on condition number $\kappa$.

We note step (1) follows directly from Theorem 8 in Zhang & Xiao (2019b). In the following, we prove step (2). First note that as $\tilde{g}_i(\boldsymbol{x}^*) \leq 0$ for all $1 \leq i \leq m$, we have:

$$\Psi(\boldsymbol{x}^{(*)}) = r(\boldsymbol{x}^{(*)}) + \gamma \ln\left(\frac{1}{m+1}\left[1 + \sum_{i=1}^{m} \exp\left(\frac{\alpha \tilde{g}_i(\boldsymbol{x}^{(*)})}{\gamma}\right)\right]\right)$$

$$\overset{\tilde{g}_i(\boldsymbol{x}^{(*)}) \leq 0}{\leq} r(\boldsymbol{x}^{(*)}) + \gamma \ln\left(\frac{1}{m+1}\left[1 + \sum_{i=1}^{m} (1)\right]\right)$$

$$= r(\boldsymbol{x}^{(*)}) \tag{23}$$

This leads to

$$\Psi(\boldsymbol{x}^{(*)}) \leq r(\boldsymbol{x}^{(*)}) + \gamma \ln(m+1) \tag{24}$$

Therefore, using item (1) we have:

$$\mathbb{E}\left[\bar{\Psi}(\boldsymbol{x}^{(K)})\right] \leq r(\boldsymbol{x}^{(*)}) + O\left(\exp(-aK)\right) \tag{25}$$

On the other hand, due to the definition of $\Psi(\boldsymbol{x})$ and smoothed max or log-sum property, we have:

$$\bar{\Psi}(\boldsymbol{x}^{(K)}) \geq r(\boldsymbol{x}^{(K)}) + \max_{1 \leq i \leq m}\left(0, \alpha \tilde{g}_i(\boldsymbol{x}^{(K)})\right). \tag{26}$$

For the purpose of lower bounding the second term in Eq. (26) we need the following Lemma:

**Lemma F.1.** *If $\bar{\boldsymbol{x}}^T \neq \boldsymbol{x}^T$, by defining $g(\boldsymbol{x}^{(K)}) \triangleq \max_{1 \leq i \leq m} \tilde{g}_i(\boldsymbol{x}^{(K)})$ for all $i \in [1:m]$ we have:*

$$g(\boldsymbol{x}^{(K)}) \geq \rho \left\|\boldsymbol{x}^{(K)} - \bar{\boldsymbol{x}}^{(K)}\right\|_2 \tag{27}$$

The proof of this lemma can be found in Mahdavi et al. (2012) but for the sake of completeness we also include the proof.

*Proof.* As $\bar{\boldsymbol{x}}^{(K)}$ is the projection of $\boldsymbol{x}^{(K)}$ into $\mathcal{K}$; i.e., $\bar{\boldsymbol{x}}^{(K)} = \underset{g(\boldsymbol{x}^{(K)}) \leq 0}{\arg\min} \left\|\boldsymbol{x} - \boldsymbol{x}^{(K)}\right\|^2$, then due to first order optimality condition, there exists a positive constant $k > 0$ such that

$$g(\bar{\boldsymbol{x}}^{(K)}) = 0 \quad \text{s.t.,} \quad \bar{\boldsymbol{x}}^{(K)} - \boldsymbol{x}^{(K)} = k \nabla g(\bar{\boldsymbol{x}}^{(K)}) \tag{28}$$

As a result we have:

$$g(\boldsymbol{x}^{(K)}) = g(\boldsymbol{x}^{(K)}) - g(\bar{\boldsymbol{x}}^{(K)}) \geq \left(\boldsymbol{x}^{(K)} - \bar{\boldsymbol{x}}^{(K)}\right) \nabla g(\bar{\boldsymbol{x}}^{(K)}) = \left\|\left(\boldsymbol{x}^{(K)} - \bar{\boldsymbol{x}}^{(K)}\right)\right\| \left\|\nabla g(\bar{\boldsymbol{x}}^{(K)})\right\|$$

$$\overset{(\oslash)}{\geq} \rho \left\|\left(\boldsymbol{x}^{(K)} - \bar{\boldsymbol{x}}^{(K)}\right)\right\| \tag{29}$$

where ($\oslash$) follows from Assumption 3. $\qquad\square$

Hence, we have:

$$\bar{\Psi}(\boldsymbol{x}^{(K)}) \geq r(\boldsymbol{x}^{(K)}) + \alpha\rho \left\|\left(\boldsymbol{x}^{(K)} - \bar{\boldsymbol{x}}^{(K)}\right)\right\| \tag{30}$$

Moreover, note that we have:

$$r(\boldsymbol{x}^{(K)}) \leq r(\boldsymbol{x}^{(*)}) + \gamma \ln(m+1) + O\left(\exp(-aK)\right) - \max_{1\leq i\leq m}\left(0, \alpha\tilde{g}_i(\boldsymbol{x}^{(K)})\right)$$

$$\leq r(\boldsymbol{x}^{(*)}) + \gamma \ln(m+1) + O\left(\exp(-aK)\right)$$

Next, we can write:

$$r(\boldsymbol{x}^{(*)}) - r(\bar{\boldsymbol{x}}^{(K)}) = r(\boldsymbol{x}^{(*)}) - r(\boldsymbol{x}^{(K)}) + r(\boldsymbol{x}^{(K)}) - r(\bar{\boldsymbol{x}}^{(K)})$$

$$\leq r(\boldsymbol{x}^{(K)}) - r(\bar{\boldsymbol{x}}^{(K)})$$

$$\leq G \left\|\left(\boldsymbol{x}^{(K)} - \bar{\boldsymbol{x}}^{(K)}\right)\right\| \tag{31}$$

Eqs. (24), (30) and (31) lead to the bound:

$$\alpha\rho \left\|\left(\boldsymbol{x}^{(K)} - \bar{\boldsymbol{x}}^{(K)}\right)\right\| \leq r(\boldsymbol{x}^{(*)}) - r(\boldsymbol{x}^{(K)}) + \gamma \ln(m+1) + O\left(\exp(-aK)\right)$$

$$\leq G \left\|\left(\boldsymbol{x}^{(K)} - \bar{\boldsymbol{x}}^{(K)}\right)\right\| + \gamma \ln(m+1) + O\left(\exp(-aK)\right) \tag{32}$$

which allows us to the following:

$$\left\|\boldsymbol{x}^{(K)} - \bar{\boldsymbol{x}}^{(K)}\right\| \leq \frac{\gamma\ln m}{\alpha\rho - G} + \frac{1}{\alpha\rho - G}O\left(\exp(-aK)\right) \tag{33}$$

Finally, we have:

$$r(\bar{\boldsymbol{x}}^{(K)}) = r(\bar{\boldsymbol{x}}^{(K)}) - r(\boldsymbol{x}^{(K)}) + r(\boldsymbol{x}^{(K)})$$

$$\leq G \left\|\boldsymbol{x}^{(K)} - \bar{\boldsymbol{x}}^{(K)}\right\| + r(\boldsymbol{x}^{(K)})$$

$$\leq G \underbrace{\left[\frac{\gamma\ln(m+1)}{\alpha\rho - G} + \frac{1}{\alpha\rho - G}O\left(\exp(-aK)\right)\right] + \gamma\ln(m+1)}_{\text{Cost of violating constraints}} + r(\boldsymbol{x}^{(*)}) + O\left(\exp(-aK)\right) \tag{34}$$

Therefore, by setting $\gamma = \frac{\exp(-aK)}{\ln(m+1)}$ we achieve the desired result.

**Proof of Corollary 4.2:** The proof follows directly from Eq. (34).

**Proof of Theorem D.1:** The proof follows directly from Theorem 8 in Zhang & Xiao (2019b).

### F.1    COMPUTATIONAL COMPLEXITY OF THEOREM 4.1 AND THEOREM D.1

Given the `GCIVR` algorithm, the sample complexity for Theorem 4.1 (where $S_t$, $B_t$ and $\tau_t$ are fixed) is

$$O\left(\max\{TB, 2T\tau S\}\right) \times K = O\left(\max\{m, \frac{5}{\sqrt{m}\mu\eta}m\} + \max\{2\frac{5}{\sqrt{m}\mu\eta}\sqrt{m}\sqrt{m}, \sqrt{m}\sqrt{m}\}\ln(1/\epsilon)\right)$$

$$= O\left(\left(\max\{m, \frac{5}{\sqrt{m}\mu\eta}m\} + \max\{\frac{10}{\mu\eta}\sqrt{m}, m\}\right)\ln(1/\epsilon)\right)$$

$$= O\left(\left[m + \kappa\sqrt{m}\right]\ln(1/\epsilon)\right) \tag{35}$$

**Proof of Theorem 4.3:** The proof follows directly from Theorem 4 in Zhang & Xiao (2019b).

## F.2 COMPUTATIONAL COMPLEXITY OF STRONGLY-CONVEX OBJECTIVES CORRESPONDING TO THEOREMS 4.3

From Theorem 4 in Zhang & Xiao (2019b) for non-convex objectives we have:

$$\mathbb{E}\left[\left\|\mathcal{G}_\eta(\bar{\boldsymbol{x}}^{(T)})\right\|^2\right] \leq \begin{cases} O\left(\frac{\ln T}{T^2}\right) & \text{if } T \leq T_0, \\ O\left(\frac{\ln m}{\sqrt{m}(T-T_0+1)}\right) & \text{else } T > T_0, \end{cases} \tag{36}$$

Considering the `GCIVR` algorithm and the bound in Eq. (36), the sample complexity for Theorem 4.3 (where $S_t$ and $\tau_t$ are fixed) can be expressed as follows:

$$\begin{cases} \sum_{t=0}^T (B_t + S_t\tau_t) & \text{if } T = O\left(\epsilon^{-1/2}\right) \leq T_0 = O\left(\sqrt{m}\right), \\ T \times (B + S\tau) & T = O\left(1/\sqrt{m}\epsilon\right) > T_0 = O\left(\sqrt{m}\right), \end{cases}$$

$$= \begin{cases} \sum_{t=0}^T 2\left(\gamma t + \beta\right)^2 & \text{if } T = O\left(\epsilon^{-1/2}\right) \leq T_0 = O\left(\sqrt{m}\right), \\ \frac{1}{\sqrt{m}\epsilon} \times (m + \sqrt{m}\sqrt{m}) & T = O\left(1/\sqrt{m}\epsilon\right) > T_0 = O\left(\sqrt{m}\right), \end{cases}$$

$$= \begin{cases} O\left(\gamma^2 T^3\right) & \text{if } T = O\left(\epsilon^{-1/2}\right) \leq T_0 = O\left(\sqrt{m}\right), \\ O\left(\frac{\sqrt{m}}{\epsilon}\right) & T = O\left(1/\sqrt{m}\epsilon\right) > T_0 = O\left(\sqrt{m}\right), \end{cases}$$

$$= \begin{cases} O\left(\epsilon^{-3/2}\right) & \text{if } T = O\left(\epsilon^{-1/2}\right) \leq T_0 = O\left(\sqrt{m}\right), \\ O\left(\frac{\sqrt{m}}{\epsilon}\right) & T = O\left(1/\sqrt{m}\epsilon\right) > T_0 = O\left(\sqrt{m}\right), \end{cases} \tag{37}$$

Therefore, depending on how large $\sqrt{m}$ is and also desired accuracy level $\epsilon$, we can decide to choose an adaptive or a non-adaptive approach.

## F.3 CONVERGENCE ANALYSIS

In this section, we extend assumptions used for centralized setting to distributed counterpart as follows:

**Assumption 6.** *We have the following assumptions:*

1) $\|f'(\boldsymbol{x}_1) - f'(\boldsymbol{x}_2)\| \leq L_h \|\boldsymbol{x}_1 - \boldsymbol{x}_2\|$ *where* $\boldsymbol{x}_1, \boldsymbol{x}_2 \in \mathbb{R}$. *We also assume that* $\|f(\boldsymbol{x}_1) - f(\boldsymbol{x}_2)\| \leq \ell_h \|\boldsymbol{x}_1 - \boldsymbol{x}_2\|$
2) $\|\nabla h_j(\boldsymbol{x}_1) - \nabla h_j(\boldsymbol{x}_2)\| \leq L_h \|\boldsymbol{x}_1 - \boldsymbol{x}_2\|$ *where* $\boldsymbol{x}_1, \boldsymbol{x}_2 \in \mathbb{R}^d$. *We also assume that* $\|h_j(\boldsymbol{x}_1) - h_j(\boldsymbol{x}_2)\| \leq \ell_h \|\boldsymbol{x}_1 - \boldsymbol{x}_2\|$
3) $\|\nabla g_j(\boldsymbol{x}_1; \xi) - \nabla g_j(\boldsymbol{x}_2; \xi)\| \leq L_g \|\boldsymbol{x}_1 - \boldsymbol{x}_2\|$ *and* $\|g_j(\boldsymbol{x}_1; \xi) - g_j(\boldsymbol{x}_2; \xi)\| \leq \ell_g \|\boldsymbol{x}_1 - \boldsymbol{x}_2\|$ *for* $\forall \boldsymbol{x}_1, \boldsymbol{x}_2 \in \mathbb{R}^d$ *and* $1 \leq j \leq p$.
4) *We suppose that* $\Psi(\boldsymbol{x})$ *in Eq. (2) is bounded from below that* $\Psi^* = \inf_{\boldsymbol{x}} \Psi(\boldsymbol{x}) > -\infty$.
5) *We assume* $r(\boldsymbol{x}) \in \mathbb{R} \cup \{\infty\}$ *is a convex and lower-semicontinues function.*

**Assumption 7** (Unbiased estimation)**.** *The mini-batch sampling is unbiased that is* $\mathbb{E}[g_j(\boldsymbol{x}; \xi)] = g_j(\boldsymbol{x})$ *and* $\mathbb{E}[\nabla g_j(\boldsymbol{x}; \xi)] = \nabla g_j(\boldsymbol{x})$ *for* $1 \leq j \leq p$.

**Convergence Analysis for Strongly Convex Objectives:**

**Assumption 8.** *We additionally for the DRO problem with Wessterstein metric suppose that the function* $r(\boldsymbol{x})$ *is smooth with module* $G$.

**Theorem F.2.** *Under Assumptions 3 and 6 to 8, when* $\Psi$ *is* $\mu$*-optimally strongly convex, if we set* $\tau_t = S_t = \sqrt{B_t} = \sqrt{m}$ *and* $T = \frac{5}{\sqrt{m}\mu\eta}$, *and* $\gamma = \frac{\exp(-K)}{\ln(m+1)}$ *letting* $\eta < \frac{2}{L_\Phi + \sqrt{L_\Phi^2 + 36G_0}}$ *and* $\alpha > \frac{G}{\rho}$, *if we let* $\boldsymbol{x}^{(k+1)} = \texttt{DistBCO}\left(\boldsymbol{x}^{(k)}\right)$ *for* $k = 0, \dots, K-1$ *using Algorithm 2 after* $K = \ln(1/\epsilon)$ *stages and returning the final solution after projecting onto the constraint set* $\mathcal{K} = \{\boldsymbol{x} | g_i(\boldsymbol{x}) \leq 0, i = 1, 2, \dots, m\}$, *i.e.,* $\bar{\boldsymbol{x}}^{(K)} = \Pi_{\mathcal{K}}(\boldsymbol{x}^{(K)})$, *solves the optimization problem in Eq. (3) with convergence error* $r(\bar{\boldsymbol{x}}^{(T)}) - r(\boldsymbol{x}^{(*)}) \leq \epsilon$ *with*

$$O\left(\left(m + \frac{m}{p} + \kappa\sqrt{m}\right)\ln\frac{1}{\epsilon}\right) \tag{38}$$

*per device number of constraint checks.*

**Remark 5** (Computational Complexity). *The sample complexity for Theorem F.2 (where $S_t$, $B_t$ and $\tau_t$ are fixed) is*

$$
\begin{aligned}
O\left(\max\{TB, 2T\tau S\}\right) \times K) &= O\left(\max\{\frac{m}{p}, \frac{5}{\sqrt{m}\mu\eta}m\} + \max\{2\frac{5}{\sqrt{m}\mu\eta}\sqrt{m}\sqrt{m}, \sqrt{m}\sqrt{m}\}\ln(1/\epsilon)\right) \\
&= O\left(\left(\max\{m, \frac{5}{\sqrt{m}\mu\eta}m\} + \max\{\frac{10}{\mu\eta}\sqrt{m}, m\}\right)\ln(1/\epsilon)\right) \\
&= O\left(\left[m + \frac{m}{p} + \kappa\sqrt{m}\right]\ln(1/\epsilon)\right)
\end{aligned}
\tag{39}
$$

**Convergence Analysis for Non-Convex Objectives:** For the non-convex case we also need the following extra assumption.

**Assumption 9** (Bounded variance). *The mini-batch sampling has bounded variance that is*

$$
\mathbb{E}\left[\|\nabla g_j(\boldsymbol{x}; \mathcal{Z}) - \nabla g_j(\boldsymbol{x})\|^2\right] \leq \sigma^2
$$

*for $1 \leq j \leq m$.*

**Theorem F.3.** *Under Assumptions 6 to 9 using Algorithm 2, for some positive constants $\beta$ and $0 \leq \zeta < \sqrt{m}$; denoting $T_0 \triangleq \frac{\sqrt{m}-\zeta}{\beta} = O\left(\sqrt{m}\right)$, if $t \leq T_0$, parameters are chosen to be $\tau_t = S_t = \sqrt{B_t} = \beta t + \zeta$; and when $t > T_0$, set $\tau_t = S_t = \sqrt{m}$. Then, if $\eta < \frac{4}{L_\Phi + \sqrt{L_\Phi^2 + 12G_0}}$ it holds that*

$$
\mathbb{E}\left[\left\|\mathcal{G}_\eta(\bar{\boldsymbol{x}}^{(T)})\right\|^2\right] \leq \begin{cases} O\left(\frac{\ln T}{T^2}\right) & \text{if } T \leq T_0, \\ O\left(\frac{\ln p}{\sqrt{m}(T-T_0+1)}\right) & T > T_0, \end{cases}
\tag{40}
$$

*This leads to the per device sample complexity of $\tilde{O}\left(\min\{\frac{\sqrt{m}}{p\epsilon} + \frac{\sqrt{m}}{\epsilon}, \frac{1}{p\epsilon^{1.5}} + \frac{1}{\epsilon^{1.5}}\}\right)$ where $\tilde{O}(.)$ notation hides logarithmic factors. Therefore, depending on the objective function $\Psi$ this sample complexity corresponds to both DRO problem with $\chi^2$ and regularized KL metrics.*

We highlight that while in distributed setting we can not reduce computational complexity in terms of order of magnitude, we can reduce computational complexity partially linearly with number of devices.

## G   PROOF OF THE DISTRIBUTED ALGORITHM

For the convince, we define the following notations

$$\boldsymbol{\xi}^{(t)} = \{\boldsymbol{\xi}_1^{(t)}, \ldots, \boldsymbol{\xi}_p^{(t)}\},$$

to denote the set of local solutions and sampled mini-batches at iteration $t$ at different machines, respectively.

We use notation $\mathbb{E}[\cdot]$ to denote the conditional expectation $\mathbb{E}_{\boldsymbol{\xi}^{(t)}}[\cdot]$.

### G.1   GENERAL LEMMAS

Before proceeding to the proof of main theorems, we first review a few properties of our algorithm that will be useful in our convergence proof:

$$\mathbb{E}\left[\boldsymbol{y}_j^{(t,i)}|\boldsymbol{x}_j^{(t)}\right] = \boldsymbol{y}_{j-1}^{(t,i)} + g_i(\boldsymbol{x}_j^{(t)}) - g_i(\boldsymbol{x}_{j-1}^{(t)})$$

$$\mathbb{E}\left[\boldsymbol{z}_j^{(t,i)}|\boldsymbol{x}_j^{(t)}\right] = \boldsymbol{z}_{j-1}^{(t,i)} + \nabla g_i(\boldsymbol{x}_j^{(t)}) - \nabla g_i(\boldsymbol{x}_{j-1}^{(t)})$$

$$\mathbb{E}\left[\boldsymbol{w}_j^{(t,i)}|\boldsymbol{x}_j^{(t)}\right] = \boldsymbol{w}_{j-1}^{(t,i)} + \nabla h_i(\boldsymbol{x}_j^{(t)}) - \nabla h_i(\boldsymbol{x}_{j-1}^{(t)}) \tag{41}$$

which due to linearity and taking average over the models of all devices, leads to

$$\mathbb{E}\left[\boldsymbol{y}_j^{(t)}|\boldsymbol{x}_j^{(t)}\right] = \boldsymbol{y}_{j-1}^{(t)} + g(\boldsymbol{x}_j^{(t)}) - g(\boldsymbol{x}_{j-1}^{(t)}) \tag{42}$$

$$\mathbb{E}\left[\boldsymbol{z}_j^{(t)}|\boldsymbol{x}_j^{(t)}\right] = \boldsymbol{z}_{j-1}^{(t)} + \nabla g(\boldsymbol{x}_j^{(t)}) - \nabla g(\boldsymbol{x}_{j-1}^{(t)})$$

$$\mathbb{E}\left[\boldsymbol{w}_j^{(t)}|\boldsymbol{x}_j^{(t)}\right] = \boldsymbol{w}_{j-1}^{(t)} + \nabla h(\boldsymbol{x}_j^{(t)}) - \nabla h(\boldsymbol{x}_{j-1}^{(t)}) \tag{43}$$

Additionally, we have equivalent update rule as follows:

$$\boldsymbol{y}_j^{(t)} = \boldsymbol{y}_{j-1}^{(t)} + \frac{1}{S^{(t)}} \sum_{\xi \in \mathcal{S}_j^{(t)}} \left[g(\boldsymbol{x}_j^{(t)};\xi) - g(\boldsymbol{x}_{j-1}^{(t)};\xi)\right] \tag{44}$$

$$\boldsymbol{z}_j^{(t)} = \boldsymbol{z}_{j-1}^{(t)} + \frac{1}{S^{(t)}} \sum_{\xi \in \mathcal{S}_j^{(t)}} \left[\nabla g(\boldsymbol{x}_j^{(t)};\xi) - \nabla g(\boldsymbol{x}_{j-1}^{(t)};\xi)\right]$$

$$\boldsymbol{w}_j^{(t)} = \boldsymbol{w}_{j-1}^{(t)} + \frac{1}{S^{(t)}} \sum_{\xi in \mathcal{S}_j^{(t)}} \left[\nabla h(\boldsymbol{x}_j^{(t)};\xi) - \nabla h(\boldsymbol{x}_{j-1}^{(t)};\xi)\right] \tag{45}$$

**Lemma G.1.** *For any* $1 \le j \le \tau_t$ *we have:*

$$\mathbb{E}\left[\left\|\boldsymbol{y}_j^{(t)} - g\left(\boldsymbol{x}_j^{(t)}\right)\right\|^2\right] \le \mathbb{E}\left[\left\|\boldsymbol{y}_0^{(t)} - g\left(\boldsymbol{x}_0^{(t)}\right)\right\|^2\right] + \sum_{r=1}^{j} \frac{\ell_g^2}{S^{(t)}} \mathbb{E}\left[\left\|\boldsymbol{x}_r^{(t)} - \boldsymbol{x}_{r-1}^{(t)}\right\|^2\right] \tag{46}$$

$$\mathbb{E}\left[\left\|\boldsymbol{z}_j^{(t)} - g'\left(\boldsymbol{x}_j^{(t)}\right)\right\|^2\right] \le \mathbb{E}\left[\left\|\boldsymbol{z}_0^{(t)} - g'\left(\boldsymbol{x}_0^{(t)}\right)\right\|^2\right] + \sum_{r=1}^{j} \frac{L_g^2}{S^{(t)}} \mathbb{E}\left[\left\|\boldsymbol{x}_r^{(t)} - \boldsymbol{x}_r^{(t)}\right\|^2\right]$$

$$\mathbb{E}\left[\left\|\boldsymbol{w}_j^{(t)} - h'\left(\boldsymbol{x}_j^{(t)}\right)\right\|^2\right] \le \mathbb{E}\left[\left\|\boldsymbol{w}_0^{(t)} - h'\left(\boldsymbol{x}_0^{(t)}\right)\right\|^2\right] + \sum_{r=1}^{j} \frac{L_h^2}{S^{(t)}} \mathbb{E}\left[\left\|\boldsymbol{x}_r^{(t)} - \boldsymbol{x}_r^{(t)}\right\|^2\right] \tag{47}$$

We note that Lemma G.1 is the generalization of Lemma 1 in Zhang & Xiao (2019b), and we provide the proof for the sake of completeness.

*Proof.* Our proof can be considered as a generalization of the proof in Zhang & Xiao (2019b) to distributed setting. For this end, we use the following equation for every fix vector $\boldsymbol{u}$:

$$\mathbf{Var}(\boldsymbol{x}) = \mathbb{E}\left[\|\boldsymbol{x} - \boldsymbol{u}\|^2\right] - \|\mathbb{E}[\boldsymbol{x}] - \boldsymbol{u}\|^2 \tag{48}$$

Therefore, letting $\boldsymbol{u} = g(\boldsymbol{x}_j^{(t)})$ we obtain:

$$\mathbb{E}\left[\left\|\boldsymbol{y}_j^{(t)} - g(\boldsymbol{x}_j^{(t)})\right\|^2 \Big| \boldsymbol{x}_j^{(t)}\right] = \left\|\mathbb{E}\left[\boldsymbol{y}_j^{(t)}\Big|\boldsymbol{x}_j^{(t)}\right] - g(\boldsymbol{x}_j^{(t)})\right\|^2 + \mathbf{Var}\left(\boldsymbol{y}_j^{(t)}\Big|\boldsymbol{x}_j^{(t)}\right)$$

$$\overset{(42)}{=} \left\|\mathbb{E}\left[\boldsymbol{y}_{j-1}^{(t)}\Big|\boldsymbol{x}_{j-1}^{(t)}\right] - g(\boldsymbol{x}_{j-1}^{(t)})\right\|^2 + \mathbf{Var}\left(\boldsymbol{y}_j^{(t)}\Big|\boldsymbol{x}_j^{(t)}\right) \qquad (49)$$

The final step of proof involves bounding the term $\mathbf{Var}\left(\boldsymbol{y}_j^{(t)}\Big|\boldsymbol{x}_j^{(t)}\right)$ as follows:

$$\mathbf{Var}\left(\boldsymbol{y}_j^{(t)}\Big|\boldsymbol{x}_j^{(t)}\right) = \mathbf{Var}\left(\boldsymbol{y}_{j-1}^{(t)} + \frac{1}{S^{(t)}} \sum_{\xi \in \mathcal{S}_j^{(t)}} \left[g(\boldsymbol{x}_j^{(t)};\xi) - g(\boldsymbol{x}_{j-1}^{(t)};\xi)\right] \Big| \boldsymbol{x}_j^{(t)}\right)$$

$$= \frac{1}{S^{(t)}}\mathbf{Var}\left(g(\boldsymbol{x}_j^{(t)};\xi) - g(\boldsymbol{x}_{j-1}^{(t)};\xi)\Big|\boldsymbol{x}_j^{(t)}\right)$$

$$\overset{(\text{✏})}{\leq} \frac{1}{S^{(t)}}\mathbb{E}\left[\left\|g(\boldsymbol{x}_j^{(t)};\xi) - g(\boldsymbol{x}_{j-1}^{(t)};\xi)\right\|^2 \Big|\boldsymbol{x}_j^{(t)}\right]$$

$$\overset{(\text{✐})}{\leq} \frac{\ell_g^2}{S^{(t)}}\mathbb{E}\left[\left\|\boldsymbol{x}_j^{(t)} - \boldsymbol{x}_{j-1}^{(t)}\right\|^2 \Big|\boldsymbol{x}_j^{(t)}\right] \qquad (50)$$

where (✏) and (✐) follows from the definition of $\mathbf{Var}(.)$ and Assumption 6.

The proof for Eq. (47) follows similarly.

The rest of the proof is similar to Zhang & Xiao (2019b) but for the sake of completeness we add the rest of the proof. For this purpose, we use the notation $\Phi(\boldsymbol{x}) \triangleq h(\boldsymbol{x}) + f(g(\boldsymbol{x}))$ and $\tilde{\nabla}\Phi(\boldsymbol{x}_j^{(t)}) = \left(\boldsymbol{z}_j^{(t)}\right)^T f'(\boldsymbol{y}_j^{(t)}) + \boldsymbol{w}_j^{(t)}$

**Lemma G.2.** *Under Assumption 6 we have:*

$$\mathbb{E}\left[\left\|\nabla\Phi(\boldsymbol{x}_j^{(t)}) - \tilde{\nabla}\Phi(\boldsymbol{x}_j^{(t)})\right\|^2\right] \leq \frac{G_0}{S^{(t)}}\sum\mathbb{E}\left[\left\|\boldsymbol{x}^{(t)} - \boldsymbol{x}^{(t)}\right\|^2\right] + \frac{\sigma_0^2}{B^{(t)}} \qquad (51)$$

*with definitions $G_0 \triangleq 3\left(g_g^4 L_f^2 + g_f^2 L_g^2\right)$ and $\sigma_0^2 \triangleq 3\left(g_g^2 L_f^2 \sigma_g^2 + g_f^2 \sigma_{g'}^2 + \sigma_h^2\right)$*

*Proof.* Using Assumption 6, we obtain:

$$\mathbb{E}\left[\left\|\tilde{\nabla}\Phi(\boldsymbol{x}_j^{(t)}) - \nabla\Phi(\boldsymbol{x}_j^{(t)})\right\|^2\right]$$

$$= \mathbb{E}\left[\left\|\left(\boldsymbol{z}_j^{(t)}\right)^T f'(\boldsymbol{y}_j^{(t)}) + \boldsymbol{w}_j^{(t)} - g'(\boldsymbol{x}_j^{(t)})f'(g(\boldsymbol{x}_j^{(t)})) - h'(\boldsymbol{x}_j^{(t)})\right\|^2\right]$$

$$= \mathbb{E}\left[\left\|\left(\boldsymbol{z}_j^{(t)}\right)^T f'(\boldsymbol{y}_j^{(t)}) - g'(\boldsymbol{x}_j^{(t)})f'(\boldsymbol{y}_j^{(t)}) + g'(\boldsymbol{x}_j^{(t)})f'(\boldsymbol{y}_j^{(t)}) - g'(\boldsymbol{x}_j^{(t)})f'(g(\boldsymbol{x}_j^{(t)})) + \boldsymbol{w}_j^{(t)} - h'(\boldsymbol{x}_j^{(t)})\right\|^2\right]$$

$$\leq 3\mathbb{E}\left[\left\|\left(\boldsymbol{z}_j^{(t)}\right)^T f'(\boldsymbol{y}_j^{(t)}) - g'(\boldsymbol{x}_j^{(t)})f'(\boldsymbol{y}_j^{(t)})\right\|^2\right] + 3\mathbb{E}\left[\left\|g'(\boldsymbol{x}_j^{(t)})f'(\boldsymbol{y}_j^{(t)}) - g'(\boldsymbol{x}_j^{(t)})f'(g(\boldsymbol{x}_j^{(t)}))\right\|^2\right]$$

$$+ 3\mathbb{E}\left[\left\|\boldsymbol{w}_j^{(t)} - h'(\boldsymbol{x}_j^{(t)})\right\|^2\right]$$

$$\overset{\text{Assumption 6}}{\leq} 3g_f^2\mathbb{E}\left[\left\|\boldsymbol{z}_j^{(t)} - g(\boldsymbol{x}_j^{(t)})\right\|^2\right] + 3g_g^2 L_f^2\mathbb{E}\left[\left\|\boldsymbol{y}_j^{(t)} - \boldsymbol{x}_j^{(t)}\right\|^2\right] + 3\mathbb{E}\left[\left\|\boldsymbol{w}_j^{(t)} - h'(\boldsymbol{x}_j^{(t)})\right\|^2\right]$$

$$(52)$$

Next step of proof is to utilize Lemma G.1 in Eq. (52), which leads to

$$
\mathbb{E}\left[\left\|\tilde{\nabla}\Phi(\boldsymbol{x}_j^{(t)}) - \nabla\Phi(\boldsymbol{x}_j^{(t)})\right\|^2\right] \leq \frac{3\left(\ell_g^4 L_f^2 + \ell_f^2 L_g^2\right)}{S^{(t)}} \sum \mathbb{E}\left[\left\|\boldsymbol{x}^{(t)} - \boldsymbol{x}^{(t)}\right\|^2\right] + 3\left(\ell_g^2 L_f^2\right)\mathbb{E}\left[\left\|\boldsymbol{y}_0^{(t)} - \boldsymbol{x}_0^{(t)}\right\|^2\right]
$$

$$
+ 3\left(\ell_f^2\right)\mathbb{E}\left[\left\|\boldsymbol{z}_0^{(t)} - \boldsymbol{x}_0^{(k)}\right\|^2\right] + \sum_{r=1}^j \frac{3L_h^2}{S^{(t)}}\mathbb{E}\left[\left\|\boldsymbol{x}_r^{(t)} - \boldsymbol{x}_r^{(t)}\right\|^2\right] + 3\mathbb{E}\left[\left\|\boldsymbol{w}_0^{(t)} - h'\left(\boldsymbol{x}_0^{(t)}\right)\right\|^2\right]
$$

$$
\overset{(a)}{=} \frac{3\left(\ell_g^4 L_f^2 + \ell_f^2 L_g^2 + L_h^2\right)}{S^{(t)}} \sum_{r=1}^j \mathbb{E}\left[\left\|\boldsymbol{x}_r^{(t)} - \boldsymbol{x}_{r-1}^{(t)}\right\|^2\right] + \frac{\sigma_0^2}{B^{(t)}}
$$

$$(53)$$

where (a) comes from algorithm and computing initial full batch in the beginning of each epoch, and using Assumption 9 as well as following bounds:

$$
\mathbb{E}\left[\left\|\boldsymbol{y}_0^{(t)} - g(\boldsymbol{x}_0^{(t)})\right\|^2\right] \leq \frac{\sigma_g^2}{B^{(t)}}, \quad \mathbb{E}\left[\left\|\boldsymbol{z}_0^{(t)} - g'(\boldsymbol{x}_0^{(k)})\right\|^2\right] \leq \frac{\sigma_{g'}^2}{B^{(t)}} \quad \text{and} \quad \mathbb{E}\left[\left\|\boldsymbol{w}_0^{(t)} - h'(\boldsymbol{x}_0^{(k)})\right\|^2\right] \leq \frac{\sigma_{h'}^2}{B^{(t)}}
$$

$$(54)$$

$\square$

The rest of the proof is to show that $\mathbb{E}\left[\left\|\mathcal{G}_\eta(\boldsymbol{x}_j^2)\right\|^2\right] \leq \epsilon$ where $\mathcal{G}_\eta(\boldsymbol{x}_j^{(t)}) \triangleq \frac{1}{\eta}\left(\boldsymbol{x}_j^{(t)} - \hat{\boldsymbol{x}}_{j+1}^{(t)}\right)$ and

$$
\hat{\boldsymbol{x}}_{j+1}^{(t)} = \Pi_f^\eta\left(\boldsymbol{x}_j^{(t)} - \eta\left[\nabla\Phi(\boldsymbol{x}_j^{(t)})\right]\right)
$$

However, Algorithm 2 produces approximate proximal gradient mapping:

$$
\tilde{\mathcal{G}}_\eta(\boldsymbol{x}_j^{(t)}) \triangleq \frac{1}{\eta}\left(\boldsymbol{x}_j^{(t)} - \boldsymbol{x}_{j+1}^{(t)}\right) \tag{55}
$$

where $\hat{\boldsymbol{x}}_{j+1}^{(t)} = \Pi_f^\eta\left(\boldsymbol{x}_j^{(t)} - \eta\nabla\Phi(\boldsymbol{x}_j^{(t)})\right)$. So the following Lemmas connect two gradient approximations:

**Lemma G.3.** *We have:*

$$
\mathbb{E}\left[\left\|\mathcal{G}_\eta(\boldsymbol{x}_j^2)\right\|^2\right] \leq 2\mathbb{E}\left[\left\|\tilde{\mathcal{G}}_\eta(\boldsymbol{x}_j^2)\right\|^2\right] + 2\mathbb{E}\left[\left\|\tilde{\nabla}\Phi(\boldsymbol{x}_j^{(t)}) - \nabla\Phi(\boldsymbol{x}_j^{(t)})\right\|^2\right] \tag{56}
$$

*Proof.* Using triangle inequality $\left\|\boldsymbol{x}_j^{(t)} - \hat{\boldsymbol{x}}_{j+1}^{(t)}\right\|^2 \leq 2\left\|\boldsymbol{x}_j^{(t)} - \boldsymbol{x}_{j+1}^{(t)}\right\|^2 + 2\left\|\boldsymbol{x}_{j+1}^{(t)} - \hat{\boldsymbol{x}}_{j+1}^{(t)}\right\|^2$ and the definition of $\tilde{\mathcal{G}}_\eta(\boldsymbol{x}_j^{(t)})$ and $\mathcal{G}_\eta(\boldsymbol{x}_j^2)$, we have

$$
\mathbb{E}\left[\left\|\mathcal{G}_\eta(\boldsymbol{x}_j^2)\right\|^2\right] \leq 2\mathbb{E}\left[\left\|\tilde{\mathcal{G}}_\eta(\boldsymbol{x}_j^2)\right\|^2\right] + \frac{2}{\eta^2}\left\|\boldsymbol{x}_j^{(t)} - \hat{\boldsymbol{x}}_{j+1}^{(t)}\right\|^2
$$

$$
= 2\mathbb{E}\left[\left\|\tilde{\mathcal{G}}_\eta(\boldsymbol{x}_j^2)\right\|^2\right] + \frac{2}{\eta^2}\left\|\Pi_f^\eta\left(\boldsymbol{x}_j^{(t)} - \eta\left[\tilde{\nabla}\Phi(\boldsymbol{x}_j^{(t)})\right]\right) - \Pi_f^\eta\left(\boldsymbol{x}_j^{(t)} - \eta\left[\nabla\Phi(\boldsymbol{x}_j^{(t)})\right]\right)\right\|^2
$$

$$
= 2\mathbb{E}\left[\left\|\tilde{\mathcal{G}}_\eta(\boldsymbol{x}_j^2)\right\|^2\right] + 2\left\|\left[\tilde{\nabla}\Phi(\boldsymbol{x}_j^{(t)}) - \nabla\Phi(\boldsymbol{x}_j^{(t)})\right]\right\|^2 \tag{57}
$$

$\square$

Based on the definition $\Psi(\boldsymbol{x}) \triangleq F(g(\boldsymbol{x})) + h(\boldsymbol{x}) + r(\boldsymbol{x})$ we have the following lemma:

**Lemma G.4.** *With definition $L_\Phi = L_F + L_h$, we have:*

$$
\mathbb{E}\left[\Psi(\boldsymbol{x}_{j+1}^{(t)})\right] \leq \mathbb{E}\left[\Psi(\boldsymbol{x}_j^{(t)})\right] - \frac{1}{2}\left(\eta - L_\Psi\eta^2\right)\mathbb{E}\left[\left\|\tilde{\mathcal{G}}_\eta(\boldsymbol{x}_j^2)\right\|^2\right] + \frac{\eta}{2}\left\|\nabla\Phi(\boldsymbol{x}_j^{(t)}) - \tilde{\nabla}\Phi(\boldsymbol{x}_j^{(t)})\right\|^2
$$

$$(58)$$

*and*

$$\mathbb{E}\left[\Psi(\boldsymbol{x}_{j+1}^{(t)})\right] \leq \mathbb{E}\left[\Psi(\boldsymbol{x}_j^{(t)})\right] - \frac{\eta}{8}\mathbb{E}\left[\left\|\tilde{\mathcal{G}}_\eta(\boldsymbol{x}_j^2)\right\|^2\right] + \frac{3\eta}{4}\mathbb{E}\left[\left\|\nabla\Phi(\boldsymbol{x}_j^{(t)}) - \tilde{\nabla}\Phi(\boldsymbol{x}_j^{(t)})\right\|^2\right]$$
$$- \left(\frac{1}{4\eta} - \frac{L_\Phi}{2}\right)\mathbb{E}\left[\left\|\boldsymbol{x}_j^{(t)} - \boldsymbol{x}_{j+1}^{(t)}\right\|^2\right] \tag{59}$$

*Proof.* Using $L_\Phi$-Lipschitz continuity of $\Psi$ and the optimality of the $\frac{1}{\eta}$–strongly convex of subproblem $(r(\boldsymbol{x}))$, we obtain

$$\Psi(\boldsymbol{x}_{j+1}^{(t)}) = F(g(\boldsymbol{x}_{j+1}^{(t)})) + h(\boldsymbol{x}_{j+1}^{(t)}) + r(\boldsymbol{x}_{j+1}^{(t)})$$
$$\leq F(g(\boldsymbol{x}_j^{(t)})) + h(\boldsymbol{x}_j^{(t)}) + \left\langle\left[\nabla F(g(\boldsymbol{x}_j^{(t)})) + \nabla h(\boldsymbol{x}_j^{(t)})\right], \boldsymbol{x}_{j+1}^{(t)} - \boldsymbol{x}_j^{(t)}\right\rangle + \frac{L_F + L_h}{2}\left\|\boldsymbol{x}_{j+1}^{(t)} - \boldsymbol{x}_j^{(t)}\right\|^2 + r(\boldsymbol{x}_{j+1}^{(t)})$$
$$= F(g(\boldsymbol{x}_j^{(t)})) + h(\boldsymbol{x}_j^{(t)}) + \left\langle\left[\tilde{\nabla} F(g(\boldsymbol{x}_j^{(t)})) + \tilde{\nabla} h(\boldsymbol{x}_j^{(t)})\right], \boldsymbol{x}_{j+1}^{(t)} - \boldsymbol{x}_j^{(t)}\right\rangle + \frac{1}{2\eta}\left\|\boldsymbol{x}_{j+1}^{(t)} - \boldsymbol{x}_j^{(t)}\right\|^2 + r(\boldsymbol{x}_{j+1}^{(t)})$$
$$+ \left\langle\left[\nabla F(g(\boldsymbol{x}_j^{(t)})) + \nabla h(\boldsymbol{x}_j^{(t)})\right] - \left[\tilde{\nabla} F(g(\boldsymbol{x}_j^{(t)})) + \tilde{\nabla} h(\boldsymbol{x}_j^{(t)})\right], \boldsymbol{x}_{j+1}^{(t)} - \boldsymbol{x}_j^{(t)}\right\rangle$$
$$+ \left(\frac{1}{2\eta} - \frac{L_F + L_h}{2}\right)\left\|\boldsymbol{x}_{j+1}^{(t)} - \boldsymbol{x}_j^{(t)}\right\|^2$$
$$\leq F(g(\boldsymbol{x}_j^{(t)})) + h(\boldsymbol{x}_j^{(t)}) + r(\boldsymbol{x}_j^{(t)}) - \frac{1}{2\eta}\left\|\boldsymbol{x}_j^{(t)} - \boldsymbol{x}_{j+1}^{(t)}\right\|^2 - \left(\frac{1}{2\eta} - \frac{L_F + L_h}{2}\right)\left\|\boldsymbol{x}_{j+1}^{(t)} - \boldsymbol{x}_j^{(t)}\right\|^2$$
$$+ \frac{\eta}{2}\left\|\left[\nabla F(g(\boldsymbol{x}_j^{(t)})) + \nabla h(\boldsymbol{x}_j^{(t)}) - \left[\tilde{\nabla} F(g(\boldsymbol{x}_j^{(t)})) + \tilde{\nabla} h(\boldsymbol{x}_j^{(t)})\right]\right]\right\|^2 + \frac{1}{2\eta}\left\|\boldsymbol{x}_{j+1}^{(t)} - \boldsymbol{x}_j^{(t)}\right\|^2$$
$$= \Psi(\boldsymbol{x}_j^{(t)}) - \frac{1}{2\eta}\left\|\boldsymbol{x}_j^{(t)} - \boldsymbol{x}_{j+1}^{(t)}\right\|^2 - \left(\frac{1}{2\eta} - \frac{L_F + L_h}{2}\right)\left\|\boldsymbol{x}_{j+1}^{(t)} - \boldsymbol{x}_j^{(t)}\right\|^2 + \frac{\eta}{2}\left\|\tilde{\nabla}\Phi(\boldsymbol{x}_j^{(t)}) - \nabla\Phi(\boldsymbol{x}_j^{(t)}))\right\|^2$$
$$\tag{60}$$

Next, we complete the proof by taking expectation from both sides of Eq. (60) concludes the proof of Eq. (58).

Next, using Lemma G.2, we have

$$-\frac{\eta}{4}\mathbb{E}\left[\left\|\tilde{\mathcal{G}}_\eta(\boldsymbol{x}_j^2)\right\|^2\right] \leq -\frac{\eta}{8}\mathbb{E}\left[\left\|\mathcal{G}_\eta(\boldsymbol{x}_j^2)\right\|^2\right] + \frac{\eta}{4}\mathbb{E}\left[\left\|\nabla\Phi(\boldsymbol{x}_j^{(t)}) - \tilde{\nabla}\Phi(\boldsymbol{x}_j^{(t)})\right\|^2\right] \tag{61}$$

We can prove Eq. (59), by simply adding Eq. (61) to both sides of Eq. (58). $\qquad\square$

We note that Lemmas G.3 and G.4 are an extension of Lemma 3 and 4 in Appendix Section of Zhang & Xiao (2019b) with difference that here the function $\Psi(\boldsymbol{x})$ includes extra function $h(\boldsymbol{x})$, which leads to $L_\Phi = L_F + L_h$ which is bigger than $L_F$ in Zhang & Xiao (2019b).

The rest of the proof is same as the proof of Theorems 4 and 8 in Zhang & Xiao (2019b) which is based on Lemmas G.3 and G.4, and for more details we refer the reader to the Appendix section of the reference Zhang & Xiao (2019b).

$\qquad\square$

# H APPROXIMATE APPROACH FOR DRO WITH WESSERSTEIN METRIC WITH NON-CONVEX OBJECTIVES

For the non-convex objectives, the optimization problem in Eq. (3) can be upper-bounded with the optimization problem as follows:

$$\inf_{\boldsymbol{x}\in\mathbb{X}}\sup_{Q\in\hat{\mathcal{D}}_N}\mathbb{E}_Q\left[h(\boldsymbol{x};\xi)\right] \leq \quad \underset{\boldsymbol{x}}{\text{minimize}} \quad r(\boldsymbol{x}) \triangleq \frac{1}{m}\sum_{i=1}^{m}f_i(\boldsymbol{x}_i) \tag{62}$$
$$\text{subject to} \quad \tilde{g}_i(\boldsymbol{x}) \leq 0, \ \forall i \in \{1, \cdots, m\}$$

We note that for the case of convex objective in optimum solution inequality holds with equality; however, in case of non-convex cost functions we do not have equality necessarily. We note that Eq. (62) with non-convex constraint can not be easily solved via augmented function approach in previous section. In order to solve the upper bound optimization problem in Eq. (62) efficiently via our suggested composite approach. For this end, we suggest to solve the following optimization problem where in constrained are modified to be convex:

$$\underset{\boldsymbol{x}}{\text{minimize}} \quad f(\boldsymbol{x}) \triangleq \frac{1}{m} \sum_{i=1}^{m} f_i(\boldsymbol{x}_i) \tag{63}$$

$$\text{subject to} \quad \hat{g}_i \triangleq \tilde{g}_i(\boldsymbol{x}) + \mu_i \left\| \boldsymbol{x} \right\|^2 \leq 0, \ \forall i \in \{1, \cdots, m\} \tag{64}$$

where we choose $\tilde{g}_i$ such that $\hat{g}_i$ are strongly convex. Therefore, we have the following relationship between optimization problem:

$$\inf_{\boldsymbol{x} \in \mathbb{X}} \sup_{Q \in \hat{\mathcal{D}}_N} \mathbb{E}_Q \left[ h(\boldsymbol{x}; \xi) \right] \leq \quad \underset{\boldsymbol{x}}{\text{minimize}} \quad r(\boldsymbol{x}) \triangleq \frac{1}{m} \sum_{i=1}^{m} f_i(\boldsymbol{x}_i) \tag{65}$$

$$\text{subject to} \quad \tilde{g}_i(\boldsymbol{x}) \leq 0, \ \forall i \in \{1, \cdots, m\}$$

$$\leq \quad \underset{\boldsymbol{x}}{\text{minimize}} \quad r(\boldsymbol{x}) \triangleq \frac{1}{m} \sum_{i=1}^{m} f_i(\boldsymbol{x}_i) \tag{66}$$

$$\text{subject to} \quad \tilde{g}_i(\boldsymbol{x}) + \mu_i \left\| \boldsymbol{x} \right\|^2 \leq 0, \ \forall i \in \{1, \cdots, m\} \tag{67}$$

