# OpenReview forum: "Learning Distributionally Robust Models at Scale via Composite Optimization"
_ICLR.cc/2022/Conference — ICLR 2022 Poster_

### Official Review · Reviewer_Q3eH · 2021-10-20

**Correctness:** 4
**Technical Novelty And Significance:** 3
**Empirical Novelty And Significance:** 4
**Recommendation:** 8
**Confidence:** 4

**Main Review:**

The paper provided a novel view of DRO as composite optimization with theoretical guarantees.  The mini-batch constraint sampling for handling heavily-constrained optimization problem is also interesting.  The experiments on the application to fairness are comprehensive and convincing.

The main weakness is that the convergence analysis heavily depends on the existing work (Zhang and Xiao, 2019b).  The only difference is the extra empirical risk term.  It is not clear that what is the main challenges in this slightly modified setting.

Other comment:  Is optimally strongly convex condition the same as PL condition?   It seems that no reference about this condition is mentioned and no specific examples are given.


**The authors have addressed my questions and the technical differences from (Zhang and Xiao, 2019b)

**Summary Of The Paper:**

The paper views different variants of DRO are simply instances of a finite-sum composite optimization, from which efficient optimization algorithms were proposed. The convergence analysis was established for strongly-convex and non-convex settings.  The effectiveness of the proposed algorithm are well demonstrated in experiments.

**Summary Of The Review:**

In summary, the paper seems to be very interesting.  It regards DRO as a composite optimization problem and proposed a a novel mini-batch constraint sampling for handling heavily-constrained optimization problems. The efficiency of the proposed algorithm is sufficiently validated by experiments. The originality and novelty of the proof techniques are very limited.

---

> ### Author Response · Authors · 2021-11-17
> **Response to Reviewer Q3eH**
>
> Thanks for your positive comments.
> ```
> The main weakness is that the convergence analysis heavily depends on the existing work Zhang and Xiao (2019b). The only difference is the extra empirical risk term. It is not clear that what is the main challenges in this slightly modified setting.
> ```
> We agree with the reviewer that the convergence analysis heavily uses the results from [Zhang and Xiao (2019b)]. Yet, we would like to highlight the main challenges in our theoretical analysis as follows: 1) For DRO problems with Wasserstein metric or heavily constrained optimization problems, we needed to show that the solution of an unconstrained finite-sum composite problem reduces to the solution of the constrained problem. 2) As discussed in the paper, regarding the lower bound, the complexity of solving DRO with our algorithm is almost optimal (in the centralized setting). Hence, we extended our results to the distributed setting (please see the Appendix section E) so that we can solve DRO problems at scale. We theoretically showed that in a distributed setting with $p$ machines we can further reduce the complexity with averaging the models.
> ```
> Other comment: Is optimally strongly convex condition the same as PL condition? It seems that no reference about this condition is mentioned and no specific examples are given.
> ```
> Thanks for catching this. According to Definition 4 of [1] optimally strong objectives or Quadratic functional growth property is the generalization of strongly convex condition. According to [2] below, the PL condition implies an optimally strongly convex condition but not vice versa. Therefore, optimally strongly convex generalizes PL condition as well. We will add these references to the subsequent version of our draft.
>
> [1]: Necoara, Ion, Yu Nesterov, and Francois Glineur. "Linear convergence of first-order methods for non-strongly convex optimization."
>
> [2]: Karimi, Hamed, Julie Nutini, and Mark Schmidt. "Linear convergence of gradient and proximal-gradient methods under the Polyak-łojasiewicz condition."

---

### Official Review · Reviewer_Wug8 · 2021-11-02

**Correctness:** 4
**Technical Novelty And Significance:** 2
**Empirical Novelty And Significance:** 2
**Recommendation:** 5
**Confidence:** 2

**Main Review:**

Strengths
1. clearly written and provide comprehensive comparison among relative works
2. The summarization of different variants of DRO are simply instances of a finite-sum composite optimization is interesting

Weaknesses:
1. While the summarization  is interesting, all of them seems already justified by other literatures.
2. The theoretically improvement of the proposed algorithm seems not that significant.

**Summary Of The Paper:**

This paper targets on solving distributionally robust optimization (DRO) that considering   distribution shifts in the data. In this paper, they show that  how different variants of DRO are simply instances of a finite-sum composite optimization for which they provide scalable methods by utilizing variance reduction algorithm. They also provide empirical results that demonstrate the effectiveness of our proposed algorithm with respect to the prior art in order to learn robust models from very large datasets.

**Summary Of The Review:**

The paper is clearly written and provide comprehensive comparison among relative works.  The summarization of different variants of DRO are simply instances of a finite-sum composite optimization is interesting. However all of them  seems already justified by other literatures. The proposed variance reduced algorithm mainly targets on solving a  finite-sum composite optimization. However, the object function involves a new ingredient  in which they also employ the variance reduction technique to solve it. As compared to the previous variance reduction work in composite optimization, the dealing with the new ingredient   may not be that difficult to handle. Therefore, I tend to reject this paper at this time.

---

> ### Author Response · Authors · 2021-11-17
> **Response to Reviewer Wug8**
>
> Thanks for your comments.
> ```
> 1. While the summarization is interesting, all of them seem already justified by other literature.
> ```
> We would like to emphasize that solving a finite-sum composite optimization problem is not the main contribution of this work. However, the reduction from  DRO to finite-sum composite optimization (as you mentioned), solving heavily constrained optimization problem via finite-sum composite optimization and extension of the results (and the algorithm) to the distributed setting (We moved the distributed setting algorithm and analysis to the appendix due to the space limitation) are the three main contributions of this work and we are not aware of any prior literature that looked at DRO problem from this point of view or to provide similar results.
>
> ```
> 2. The theoretical improvement of the proposed algorithm seems not that significant.
> ```
> As shown in Table 1, for general DRO with Wasserstein metric (or equivalently heavily constrained optimization problem),  [Cotter et al. (2016)] is the most relevant work for which we improve the complexity remarkably both in terms of dependency on the convergence error ($\epsilon$) and the number of constraints $m$,  in addition to reducing the storage cost of the probability distribution of the dimension $m$. Additionally, compared to SEVR [Yue et al (2021)], we also improve the complexity in terms of the dependency on the convergence error ($\epsilon$) even for the special case of the General Linearized Wasserstein problem (unconstrained).
>
> Regarding the DRO with $\chi^2$ or KL metrics for non-convex objectives, to the best of our knowledge, this is the first work with theoretical guarantees.
>
> Therefore, we believe our theoretical results either significantly improve upon prior literature or introduce new results.

---

### Official Review · Reviewer_rHQv · 2021-11-04

**Correctness:** 4
**Technical Novelty And Significance:** 3
**Empirical Novelty And Significance:** 2
**Recommendation:** 6
**Confidence:** 2

**Main Review:**

The paper is well written and the method is well motivated. The theoretical analysis seems to be incremental and is very similar to those in the literature such as Fang et al. and Zhang & Xiao.

Can you explain the intuition behind Assumption 3? Why is there a gradient lower bound imposed?

What is the choice of \gamma in Corollary 4.2 that makes the results match that in Theorem 4.1?

In the experimental results, the variance seems to be very high. How many repetitions are conducted to obtain the results?

Why did you only compare with one baseline algorithm in the experiments? Although the algorithm structure and the relaxation might be different, it will be good to see the empirical comparison of the proposed algorithm with other algorithms on distributionally robust optimization such as those in Table 1.


**Summary Of The Paper:**

In this paper, the authors propose to study different distributionally robust optimization problems in the same form of a composite optimization problem. They propose a variance reduction type of gradient based algorithm to solve the composite optimization problem and prove the convergence rate in both (strongly) convex and nonconvex settings.



**Summary Of The Review:**

My recommendation is mainly based on the contribution of this work in theoretical results compared with existing work.

---

> ### Author Response · Authors · 2021-11-17
> **Response to Reviewer rHQv**
>
> ```
> The paper is well written and the method is well motivated. The theoretical analysis seems to be incremental and is very similar to those in the literature such as Fang et al. and Zhang & Xiao.
>
> ```
> Thanks for your positive feedback.
>
> We agree with the reviewer that our theoretical analysis uses the tools from [Zhang & Xiao] and builds upon it. Yet, besides the novel reformulation of DRO as finite-sum composite optimization problem, we would like to highlight that one of the distinguishing features of our work (independent from the convergence analysis in [Zhang & Xiao]) is that we showed that we can efficiently (with nearly optimal complexity) solve heavily constrained optimization problems via finite-sum composite. The reduction and the results from unconstrained to constrained problem are non-trivial (Please see Section F in Appendix). Furthermore, as we discussed, in addition to providing an almost optimal complexity for the centralized setting, we extend the results of [Zhang & Xiao] to the distributed setting (we moved this section to the Appendix Section E due to page limitation).
>
>
> ```
> Can you explain the intuition behind Assumption 3? Why is there a gradient lower bound imposed?
> ```
> We note that Assumption 3 is only needed for DRO with Wasserstein metric (or heavily constrained optimization problem) which is a standard assumption used in previous work including [Cotter et al. (2016)] and [Mahdavi et al. (2012)]. Regarding the intuition for this Assumption, we can say this assumption becomes useful in Eq. (29) of Appendix F which implies a lower bound on the value of the algorithm with and without the projection step. This assumption can translate into the absence of constraint violations and does not limit the applicability of the proposed algorithm as shown in [Cotter et al. (2016)] and [Mahdavi et al. (2012)] and related studies on online learning with long term constraints.
>
> ```
> What is the choice of \gamma in Corollary 4.2 that makes the results match that in Theorem 4.1?
> ```
> As we stated in Theorem 4.1, by choosing $\gamma=\frac{\exp(-K)}{\ln(m+1)}$ we enforce the error between the solutions of constrained and unconstrained to match with the convergence error of the unconstrained finite-sum composite optimization problem.
> ```
> In the experimental results, the variance seems to be very high. How many repetitions are conducted to obtain the results?
> ```
> The experiments are repeated 3 times to get the mean results. In most experiments, the variance of all approaches including unconstrained and GCVIR seem to be normal. However, as the reviewer mentioned, the variance of the heavily constrained approach in the ranking fairness problem seems to be high. This might be due to a large number of constraints to satisfy and non-convexity of the objective. We are using the codebase provided by the authors of the algorithm and run the experiment using the suggested hyperparameters.
> ```
> Why did you only compare with one baseline algorithm in the experiments? Although the algorithm structure and the relaxation might be different, it will be good to see the empirical comparison of the proposed algorithm with other algorithms on distributionally robust optimization such as those in Table 1.
> ```
> Regarding the experiments, the other paper in Table 1 has not provided any reproducible code so we could not compare our algorithms. Additionally, as you pointed out the structure of the algorithm for the reference SEVR [Yue et al (2021)] is different than ours as they do consider a special case of general linearized Wasserstein unconstrained DRO problem. However, we consider the general Wassterstein DRO problem, which is reformulated as a heavily constrained optimization problem. Therefore, we compare our algorithm mainly with the heavy-touch algorithm in [Cotter et al. (2016)] which also targets solving heavily constrained problems.

---

### Author Response · Authors · 2021-12-07
**Post rebuttal**

We thank the reviewers for their constructive comments. In particular, we would like to thank reviewer Q3eH for finding our answers satisfactory. We very much hope that our rebuttal clarifies the issues raised by reviewers rHQv and Wug8.

---

### Decision · Program_Chairs · 2022-01-20

**Decision:**

Accept (Poster)

**Comment:**

The main concerns from the reviewers is the novelty of the algorithm and analysis from the CIVR algorithm of Zhang and Xiao (2019b). The author rebuttal clarified the main contributions as reformulation of DRO as composite finite-sum optimization, solving heavily constrained optimization problems through composite optimization, and extension to distributed algorithms. They indeed lead to meaningful contributions to the important topic of DRO and open new avenues for structured constrained optimization problems. The paper is written very clearly and the empirical results on realistic problems are much appreciated.